# ANYUP: UNIVERSAL FEATURE UPSAMPLING

**Thomas Wimmer[1,2], Prune Truong[3], Marie-Julie Rakotosaona[3], Michael Oechsle[3]**
**Federico Tombari[3,4], Bernt Schiele[1], Jan Eric Lenssen[1]**
[1]Max Planck Institute for Informatics, SIC, [2]ETH Zurich, [3]Google, [4]TU Munich
`https://wimmerth.github.io/anyup`

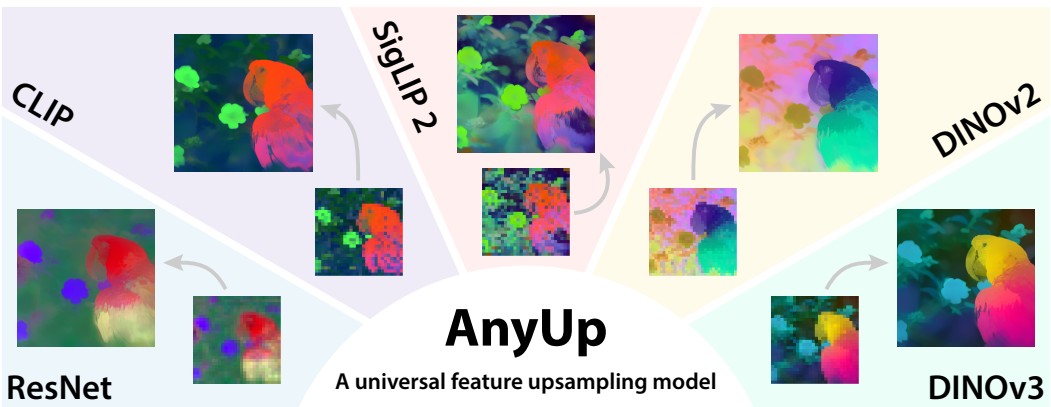

Figure 1: **AnyUp** is a universal feature upsampling model that can upsample any feature from any to any resolution, generalizing to unseen features while achieving state-of-the-art performance.

## ABSTRACT

We introduce AnyUp, a method for feature upsampling that can be applied to any vision feature at any resolution, without encoder-specific training. Existing learning-based upsamplers for features like DINO or CLIP need to be re-trained for every feature extractor and thus do not generalize to different feature types at inference time. In this work, we propose an *inference-time* feature-agnostic upsampling architecture to alleviate this limitation and improve upsampling quality. In our experiments, AnyUp sets a new state of the art for upsampled features, generalizes to different feature types, and preserves feature semantics while being efficient and easy to apply to a wide range of downstream tasks.

## 1 INTRODUCTION

General image feature extractors, such as DINO (Caron et al., 2021; Oquab et al., 2023; Xue et al., 2025), CLIP (Radford et al., 2021), SigLIP (Tschannen et al., 2025), or MAE (He et al., 2022), have become fundamental building blocks in modern day computer vision, providing, e.g., semantics or language-alignment to a wide range of down-stream applications, such as depth estimation or 3D reconstruction (Feng et al., 2024; Wang et al., 2025), open-vocabulary semantic segmentation (Engelmann et al., 2024; Wysoczańska et al., 2024), or benchmarking generative models (Asim et al., 2025). An important limitation of such pre-trained models, which are usually transformer-based, is that their output feature map resolution is limited to the number of transformer tokens, preventing the prediction of pixel-level features. Therefore, several recent works, such as FeatUp (Fu et al., 2024), LoftUp (Huang et al., 2025), or JAFAR (Couairon et al., 2025) propose learned feature upsampling methods.

While such feature upsampling methods perform well when paired with the vision encoders with which they were trained, they are generally not encoder-agnostic at inference time and need to be retrained to be usable with a different feature extractor. This can be costly or, in the case of the latest large vision models (Xue et al., 2025), even infeasible with limited computing resources,

Table 1: **Categorization of feature upsampling methods.** AnyUp is the first learnable method that generalizes to any input feature *at inference time*, while being able to upsample from any to any resolution and being task-agnostic.

| | any encoder | any resolution | any task | trainable |
|---|:---:|:---:|:---:|:---:|
| Bilinear Upsampling | ✓ | ✓ | ✓ | ✗ |
| Nearest-Neighbor Upsampling | ✓ | ✓ | ✓ | ✗ |
| Large Input Image | ✓ | ✓ | ✓ | ✗ |
| Bilateral Filtering (Tomasi & Manduchi, 1998) | ✓ | ✗ | ✓ | ✗ |
| Guided Filtering (He et al., 2012) | ✓ | ✗ | ✓ | ✗ |
| Change of ViT Patchification Stride | ✓ | ✗ | ✓ | ✗ |
| SAPA (Lu et al., 2022) | ✗ | ✗ | ✗ | ✓ |
| CARAFE (Wang et al., 2019) | ✗ | ✗ | ✗ | ✓ |
| DySample (Liu et al., 2023) | ✗ | ✗ | ✗ | ✓ |
| ReSFU (Zhou et al., 2025) | ✗ | ✓ | ✗ | ✓ |
| Resize Convolution (Odena et al., 2016) | ✗ | ✗ | ✓ | ✓ |
| LiFT (Suri et al., 2024) | ✗ | ✗ | ✓ | ✓ |
| FeatUp (Fu et al., 2024) | ✗ | ✗ | ✓ | ✓ |
| FeatSharp (Ranzinger et al., 2025) | ✗ | ✗ | ✓ | ✓ |
| LIIF (Chen et al., 2021) | ✗ | ✓ | ✓ | ✓ |
| LoftUp (Huang et al., 2025) | ✗ | ✓ | ✓ | ✓ |
| JAFAR (Couairon et al., 2025) | ✗ | ✓ | ✓ | ✓ |
| AnyUp (Ours) | ✓ | ✓ | ✓ | ✓ |

as, during training of the upsampler, the vision encoder must be queried multiple times for every training sample, often also on a higher-resolution image. In this work, we alleviate this limitation and propose *AnyUp*, a learned feature upsampling method that generalizes to features of *any size* provided in *any resolution* while achieving state-of-the-art performance across various downstream tasks. As shown in Tab. 1, AnyUp is the first of its kind.

Feature upsamplers extract information from a low-resolution feature map and a high-resolution RGB guidance image, to infer which pixel in the high-resolution image should receive which feature (Couairon et al., 2025). The key limitation of existing work is that their way of processing the low-resolution feature map is specific to the dimensionality and type of the used feature. In contrast, AnyUp employs a feature-agnostic layer that allows to process any feature and can also generalize to novel feature types. In addition to the feature-agnostic layer, we introduce a window attention procedure and a crop-based training strategy, which further improve feature upsampling quality.

In our experiments, AnyUp establishes a new state of the art for feature upsampling. Its key advantages include universal applicability – it can be trained and applied across all feature types and resolutions – and robust generalization to feature types it was not trained on. Moreover, AnyUp ensures high fidelity by minimizing the distortion of original feature semantics. Through ablations, we analyze different design choices and draw general insights for the feature upsampling task.

In summary, our contributions are:

- We introduce AnyUp, an upsampling model that is feature-agnostic *at inference time*, *i.e.*, that needs to be trained just once and can be used to upsample features from any source at any resolution and dimensionality.
- We propose a feature-agnostic layer that captures information from features of varying type and dimensionality.
- We introduce a window attention-based upsampling architecture that can be trained effectively using an image part-based loss and retains the input feature space through consistency regularization.
- We show that AnyUp outperforms prior upsampling methods while being able to generalize to any vision encoder at test time.

We make our code and pre-trained weights publicly available at https://github.com/wimmerth/anyup, giving access to a light-weight, training-free and easy-to-use feature upsampler.

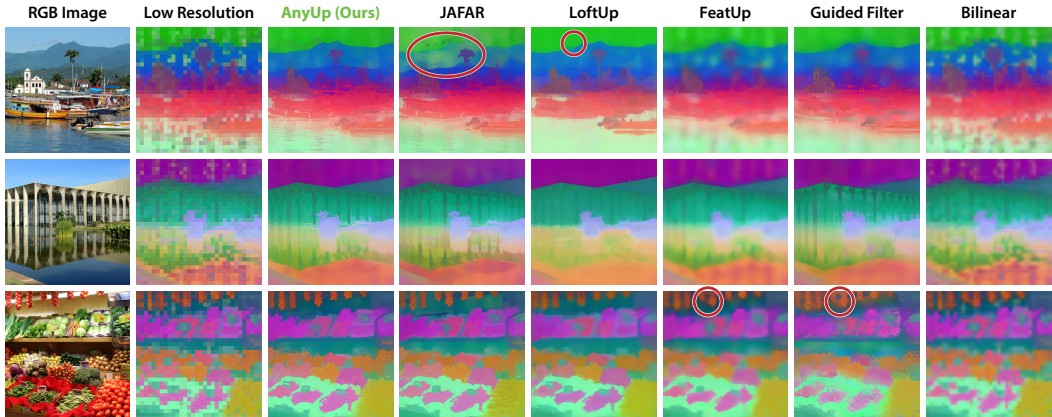

Figure 2: **Visual comparison against other methods.** RGB channels correspond to the first three principal components computed over all features. Previous methods result in excessive smoothing or contain other artifacts: See, *e.g.*, in the first row, the smoothed-out cloud features in LoftUp or the feature distribution shift for the mountains in JAFAR, as well as the oversmoothing and halo-artifacts of FeatUp and Guided Filter in the third row. AnyUp (Ours) produces sharp output feature maps while preserving the input feature quality.

## 2 RELATED WORK

Prior works on feature upsampling can be categorized into several groups, based on their capabilities and reliance on data, as summarized in Tab. 1.

Training-free upsampling methods can usually be used with any input features, for any task, and can upscale to any resolutions, while being relatively light-weight. This makes them, *i.e.*, bilinear upsampling, the de-facto standard choice in many applications where high-resolution features are required. Methods like Guided Filtering (He et al., 2012) also work surprisingly well on favorable data samples, see Fig. 2, but degrade on, *e.g.*, low-contrast or more complex images, requiring per-sample tuning of hyperparameters and often still causing excessive or insufficient blur (see the last row in Fig. 2). Changing the stride of the patchification in the vision transformer (Dosovitskiy et al., 2021), or simply increasing the resolution of the input image results in computationally heavy inference and risks moving the features out-of-distribution (Ranzinger et al., 2025).

Learnable upsampling methods (Suri et al., 2024; Fu et al., 2024; Huang et al., 2025; Couairon et al., 2025) try to improve the quality and robustness over learning-free approaches. The parameterization, however, always results in the loss of feature independence at inference time, *i.e.*, upsampling models are fitted to a specific feature encoder (and input dimensionality). In addition, some methods are fixed to a specific downstream task or model (Wang et al., 2019; Lu et al., 2022; Liu et al., 2023; Zhou et al., 2025) or cannot be used with arbitrary scaling factors (Odena et al., 2016; Wang et al., 2019; Lu et al., 2022; Liu et al., 2023; Suri et al., 2024; Fu et al., 2024; Ranzinger et al., 2025). The current state-of-the-art approaches, JAFAR (Couairon et al., 2025) and LoftUp (Huang et al., 2025), both concurrent works, formulate feature upsampling as a single or stacked high-res to low-res attention, respectively. We adopt this formulation as it offers the advantage of being naturally resolution-agnostic, *i.e.*, features can be upsampled from any to any resolution. However, in our AnyUp, we aim to make this architecture encoder-agnostic, *i.e.*, the upsampling model should be trained once and subsequently be applicable to features from any vision encoder, potentially of different feature dimensionality. More details on selected related work can be found in App. A.

## 3 TASK FORMULATION

Given an input RGB image $I_{hr} \in \mathbb{R}^{H \times W \times 3}$, the goal of feature upsampling methods $f$ is to upsample low-resolution input feature maps $p \in \mathbb{R}^{h \times w \times c} := e(I_{hr})$ to high-resolution output feature maps $q := f(p, I_{hr}) \in \mathbb{R}^{H \times W \times c}$. The necessity of this task is grounded in the use of large pre-trained vision encoders $e(\cdot)$ as feature extractors and powerful semantic priors in recent years. Almost all such models, be it convolutional neural networks (Fukushima, 1969; LeCun et al., 1989) or vi-

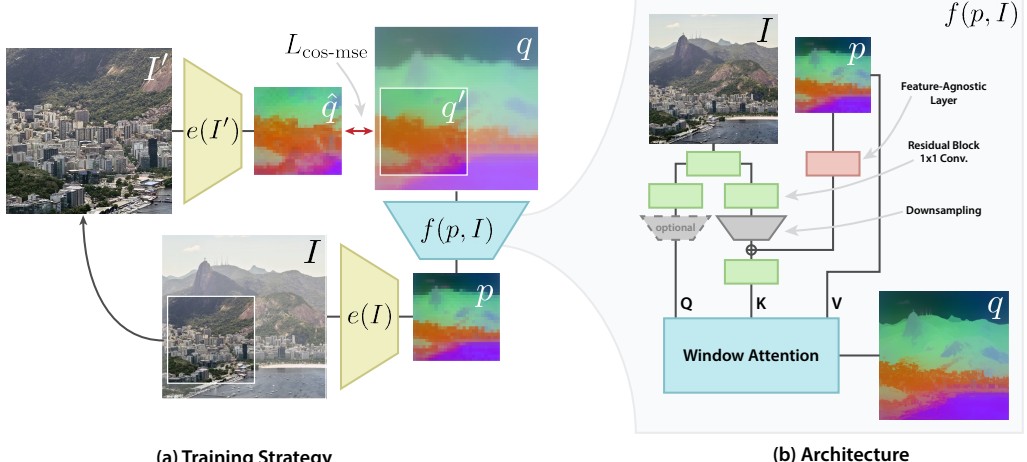

Figure 3: **Method Overview.** AnyUp performs window attention-based upsampling (4.2). Input features are processed with a feature-agnostic layer (4.1). During training, features computed for randomly sampled image parts are used as a reference for the respective part of the upsampled feature map (4.3).

sion transformers (Dosovitskiy et al., 2021), perform learned downsampling to latent resolutions $h \times w < H \times W$, which is necessary for efficient image processing using these large neural networks. However, when exploiting the features of these powerful models for downstream tasks, it is often necessary to work with high-resolution features, e.g., when performing dense, pixel-wise predictions, or aggregating information from multiple views in 3D (Engelmann et al., 2024; Ye et al., 2024; Asim et al., 2025).

## 4 LEARNING ENCODER-AGNOSTIC FEATURE UPSAMPLING

We are interested in a lightweight and low-parameterized model to prevent memory and compute bottlenecks. For this, we base our model on an attention-based architecture, similar to that in previous works (Couairon et al., 2025; Huang et al., 2025) (see Fig. 3). More specifically, we adopt the architecture of JAFAR (Couairon et al., 2025), which we briefly describe next. The input image $I$ and low-resolution feature map $p$ are both first passed through convolution blocks with residual connections. Next, positional encodings are applied to the image features. Queries for the final attention layer are computed directly from these pixel features, while information from both the downsampled image and the low-resolution feature map is used to compute the keys. The values in the attention layer are simply the unprocessed patch features from the input feature map.

In our method, we aim to fix two of the main limitations of this architecture. First, we replace the initial feature processing layer, which acts only on fixed-dimensional inputs and needs to be learned per vision backbone with our proposed feature-agnostic convolution layer (Sec. 4.1). Second, we simplify the task of the upsampler by restricting the source features to local windows in the attention computation (Sec. 4.2).

We further improve the training pipeline for our feature upsampling method by using an image part-based strategy that is explained in Sec. 4.3.1, as well as additional consistency regularization improving the robustness to noise and input feature space preservation (Sec. 4.3.2).

### 4.1 FEATURE-AGNOSTIC UPSAMPLING BY DESIGN

In our feature-agnostic layer, we aim to represent feature maps from any source model with any dimensionality as feature maps with canonical dimensionality by convolving them with a learned filter basis. We hypothesize that an attention-based upsampling model, where the outputs are linear combinations of the input features for every pixel, mostly needs to understand overall local structure changes in the input feature map. To capture this structural information while staying agnostic to the input features (and their dimensionality), we design a convolutional layer that captures structure independently for all input channels and aggregates this information later. We illustrate the structure of this layer in Fig. 4.

In our proposed feature-agnostic layer, each input channel $p_i$ is convolved with a learned kernel basis $\{\psi_j \in \mathbb{R}^{k \times k}\}_{j=1,...,M}$, followed by a softmax operation on the activations along the basis filter dimension. The resulting filter-wise contributions are then averaged over all input channels. More formally, the output feature $f_j$ of the feature-agnostic convolution layer can be computed as

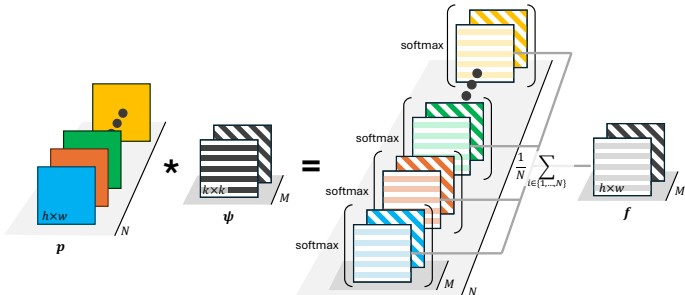

Figure 4: **Feature-agnostic layer.** Input channels are processed independently and contributions to basis filters are averaged over all channels leading to outputs invariant to input dimensionality.

$$f_j = \frac{1}{N} \sum_{i \in \{1,...,N\}} \frac{\exp(p_i * \psi_j)}{\sum_{j' \in \{1,...,M\}} \exp(p_i * \psi_{j'})}, \tag{1}$$

where $N$ is the (varying) number of input channels and $M$ the canonical number of output channels. Note that we leave out the spatial structure of the convolution operation to simplify notation in Eq. 1.

## 4.2 LOCAL WINDOW ATTENTION

Analyzing the attention patterns of JAFAR (Couairon et al., 2025), we find that in the global attention mechanism, where a pixel query can attend to any feature patch in the input feature map, sometimes vastly unrelated and distant image areas are used as references for upsampling. We argue that we can avoid such patterns and simplify the upsampling problem by restricting the attention computation to local windows around the query point (Ramachandran et al., 2019). As a high-resolution feature is now a linear combination of a much smaller set of coarse features compared with the global attention, the optimization objective for the upsampler gets easier, as well as efficiency is improved. We refer the reader to App. E for a visualization of the mentioned attention outliers.

## 4.3 TRAINING PIPELINE

Computing high-resolution features, *i.e.*, with a very high-resolution input image, is infeasible for obtaining a reference "ground-truth" that one can compare to at training time. Besides being computationally infeasible at scale, extreme high-resolution inputs also effectively move most models out-of-distribution, as noted by Ranzinger et al. (2024).

To circumvent this, prior work proposed different techniques to leverage low-resolution features. FeatUp (Fu et al., 2024) proposes upsampler training as "multi-view reconstruction", where equivariance to small image-space perturbations is optimized for, which requires carefully designing these augmentations without shifting the images out-of-distribution for the vision backbone. In LoftUp (Huang et al., 2025), segmentation masks are used as high-resolution guidance signal at the cost of having to query a large segmentation model in every training step. In the simplest case, as in JAFAR Couairon et al. (2025), training is only performed at low resolutions, *i.e.*, learning to upsample a 16x16 to 32x32 feature map, where both are computed from respectively scaled input images (Suri et al., 2024; Couairon et al., 2025). While Couairon et al. (2025) argue that this low-resolution training is sufficient for training an attention-based upsampling mechanism, we propose an improved training strategy based on local image crops that lets us train a more powerful feature upsampler, while being faster and more memory-efficient.

### 4.3.1 DATA SAMPLING

We make use of the fact that we do not necessarily need to supervise the upsampling on the full feature map. Instead, we choose to supervise our method only on smaller local crops. Our training pipeline is illustrated in Fig. 3. More specifically, we take a high-resolution image $I \in \mathbb{R}^{H \times W}$ and randomly sample a smaller local crop thereof $I' \in \mathbb{R}^{h \times w}$. We then downsample $I$ to the same

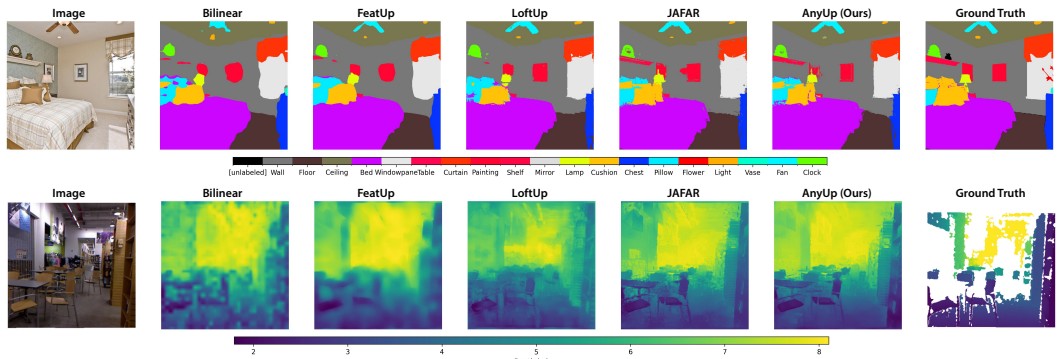

Figure 5: **Qualitative Probing Results.** Visualization of linear probing results for monocular depth estimation on NYUv2 and semantic segmentation on ADE20k. More visualizations given in App. E.

resolution $h \times w$. Using the computed features $p = e(I)$ and $\hat{q} = e(I')$, we upsample $p$ to a high-resolution feature map $q = f(I, p) \in \mathbb{R}^{h' \times w'}$, where $h', w'$ are chosen in such way that the cropped part $q'$ corresponding to $I'$ matches the resolution of $\hat{q}$. Note that while LoftUp (Huang et al., 2025, Sec. 4.2) proposes a similar strategy in their second training stage, they compute the loss against an EMA of their method instead of ground-truth features and at much higher resolution, resulting in much more compute-heavy training. On the other hand, our sampling method is more light-weight than JAFAR's simple strategy Couairon et al. (2025) described in Sec. 4.3, as we do not need to compute reference features at higher resolution, *e.g.*, 448x448.

### 4.3.2 OBJECTIVE FUNCTION

We follow prior work in feature denoising and feature upsampling (Yang et al., 2024; Couairon et al., 2025) and minimize the distance between predicted features $q'$ and target features $\hat{q}$:

$$L_{\text{cos-mse}}(q', \hat{q}) = 1 - \cos(q', \hat{q}) + L^2(q', \hat{q}). \tag{2}$$

In addition, we add a self-consistency regularization $L_{\text{self-consistency}}$, which we detail in App. C.1, as well as an input-consistency regularization $L_{\text{input-consistency}}$, where we simply compute $L_{\text{cos-mse}}$ with the input features $p$ and the accordingly downsampled predicted output features $q$. We do this to further improve the locality of the upsampled features, which is important for sub-object-level tasks like surface normal estimation, as well as maintaining the input feature space.

## 5 EXPERIMENTS

We train AnyUp on the ImageNet dataset (Krizhevsky et al., 2017) and provide further implementation details in App. C. Comparisons against previous works are provided in Sec. 5.1, where we evaluate probing quality, feature space preservation, and behaviour with different resolution changes. Sec. 5.2 provides more results on the generalization of AnyUp to other features at test time. Sec. 5.3 wraps up the experiments with an ablation study.

### 5.1 COMPARISON TO PRIOR ART

We mainly compare our method to FeatUp (Fu et al., 2024), as well as the concurrent LoftUp (Huang et al., 2025) and JAFAR (Couairon et al., 2025), which both are general, attention-based feature upsampling methods that can upsample from and to any resolution[1]. However, we note that both models have the significant disadvantage of not being agnostic to the input features at test time and need to be retrained per vision encoder. Our AnyUp does not suffer from this problem and is thus a more general method that can be applied out-of-the-box to any new features. If not mentioned otherwise, the DINOv2 ViT-S model (Oquab et al., 2023) is used as feature extractor for training and testing in the experiments, as other works also provide pre-trained weights for this model.

---

[1]While both, LoftUp and JAFAR, are considered concurrent according to ICLR guidelines, we choose to compare AnyUp mainly against these methods and FeatUp, as they are the strongest available competitors.

**Qualitative Analysis.** We visualize PCA projections of the upsampled features in Fig. 2, which shows a visual comparison against prior methods. We observe that AnyUp produces sharp output feature maps while preserving the input feature quality. Prior works, on the other hand, often still result in oversmoothed feature maps or feature distribution shifts as indicated by the red circles.

**Semantic Segmentation.** We follow the evaluation scheme proposed by prior works (Couairon et al., 2025, Sec. 4.3.1), where a linear probe, *i.e.*, a 1x1 convolution layer, is trained to predict semantic segmentation labels from the high-resolution feature maps. The input images are resized to 448x448 resolution and features are thus of resolution 28x28 or 32x32 depending on the patch size of the vision transformer used for feature extraction. Upsampling is generally performed back to the input image resolution, *i.e.*, equaling a 14x or 16x upsampling, if not mentioned otherwise. We evaluate performance on the COCO-Stuff (Lin et al., 2014), ADE20k (Zhou et al., 2018) and PASCAL VOC (Everingham et al., 2014) datasets.

Results are reported as mean Intersection-over-Union (mIoU) and pixel-wise accuracy. As observed in Tab. 2, AnyUp provides state-of-the-art upsampling performance for downstream semantic segmentation.

Table 2: **Semantic Segmentation.** Highlights for **best**, second and third best scores.

|  | COCO | | PASCAL-VOC | | ADE20k | |
|---|---|---|---|---|---|---|
|  | mIoU ($\uparrow$) | Acc. ($\uparrow$) | mIoU ($\uparrow$) | Acc. ($\uparrow$) | mIoU ($\uparrow$) | Acc. ($\uparrow$) |
| Bilinear | 59.48 | 79.32 | 81.43 | 95.38 | 40.54 | 74.12 |
| FeatUp | 61.95 | 81.14 | 83.37 | 96.04 | 42.19 | 75.57 |
| LoftUp | 62.15 | 81.32 | 83.69 | 96.11 | 42.02 | 75.72 |
| JAFAR | 61.82 | 81.07 | **84.36** | **96.22** | 42.06 | 75.48 |
| **AnyUp** | **62.16** | **81.37** | 84.00 | 96.19 | **42.43** | **75.85** |

**Depth and Normal Estimation.** We follow Probe3D (Banani et al., 2024) for evaluating the downstream performance on depth and surface normal estimation. Notably, we resize input images, target depth and surface normal maps to 224x224 resolution, if not mentioned otherwise. Results are reported as the root mean square error (RMSE). For normal estimation, we further report the accuracy of predictions for specified angular thresholds. For depth estimation, we additionally report the $\delta_1$ score, corresponding to the amount of pixels for which the ratio of prediction to ground-truth is less than $1.25$. For more details on the benchmarking, as well as the employed metrics, we refer the reader to Banani et al. (2024, App. A.3).

As shown in Tab. 3, AnyUp outperforms all competitors on these tasks, showcasing its strength in preserving the locality of upsampled features. In contrast, LoftUp smoothens the features per-object too much due to its training objective using object segmentation masks and therefore performs suboptimal on tasks like surface normal estimation.

Table 3: **Surface Normal and Monocular Depth Estimation.** AnyUp outperforms previous upsamplers for geometric tasks. Evaluation on the NYUv2 dataset (Silberman et al., 2012).

|  | Surface Normals | | | | Depth (Absolute) | | Depth (Relative) | |
|---|---|---|---|---|---|---|---|---|
|  | RMSE ($\downarrow$) | 11.25° ($\uparrow$) | 22.5° ($\uparrow$) | 30° ($\uparrow$) | RMSE ($\downarrow$) | $\delta_1$ ($\uparrow$) | RMSE ($\downarrow$) | $\delta_1$ ($\uparrow$) |
| Bilinear | 32.70 | 0.26 | 0.53 | 0.66 | 0.4925 | 0.8081 | 0.3582 | 0.9112 |
| FeatUp | 32.69 | 0.25 | 0.53 | 0.66 | 0.4816 | 0.8156 | 0.3413 | 0.9193 |
| LoftUp | 33.94 | 0.26 | 0.51 | 0.64 | 0.4847 | 0.8127 | 0.3478 | 0.9166 |
| JAFAR | 31.54 | 0.28 | 0.56 | 0.68 | 0.4906 | 0.8052 | 0.3481 | 0.9180 |
| **AnyUp** | **31.17** | **0.29** | **0.57** | **0.69** | **0.4755** | **0.8216** | **0.3378** | **0.9233** |

**Upsampling from Any to Any Resolution.** We follow the same evaluation procedure as described before but now vary the input feature resolution and target upsampling size, where the latter also corresponds to the size of the guidance image given to the upsampler methods. The semantic segmentation results are obtained on the COCO dataset. As FeatUp (Fu et al., 2024) does not support varying upsampling ratios, we always upsample by a factor of 16x and subsequently perform bilinear downsampling to the desired output resolution. Results are shown in Tab. 4. AnyUp outperforms its competitors across most resolution changes while only slightly performing worse than the best competing method when upsampling from $16 \rightarrow 112$ pixels.

Table 4: **Upsampling from any to any resolution.** Linear probing results for Semantic Segmentation (COCO) and depth estimation when varying the feature map and output resolutions.

| | Semantic Segmentation | | | | | | Depth Estimation | | | | | |
| | $16 \to 112$ | | $32 \to 224$ | | $32 \to 112$ | | $16 \to 112$ | | $32 \to 224$ | | $32 \to 112$ | |
| | mIoU (↑) | Acc. (↑) | mIoU (↑) | Acc. (↑) | mIoU (↑) | Acc. (↑) | RMSE (abs) (↓) | RMSE (rel) (↓) | RMSE (abs) (↓) | RMSE (rel) (↓) | RMSE (abs) (↓) | RMSE (rel) (↓) |
|---|---|---|---|---|---|---|---|---|---|---|---|---|
| Bilinear | 56.38 | 77.17 | 59.42 | 79.28 | 59.40 | 79.27 | 0.4927 | 0.3586 | 0.4606 | 0.3274 | 0.4600 | 0.3273 |
| FeatUp | 58.88 | 79.15 | 61.92 | 81.10 | 61.76 | 80.99 | **0.4357** | **0.3231** | 0.4507 | 0.3145 | 0.4513 | 0.3160 |
| LoftUp | 58.97 | 79.37 | 61.68 | 81.06 | 61.20 | 80.69 | 0.4896 | 0.3533 | 0.4591 | 0.3264 | 0.4636 | 0.3296 |
| JAFAR | **59.79** | **79.87** | 61.91 | 81.14 | 61.66 | 80.94 | 0.4871 | 0.3458 | 0.4825 | 0.3489 | 0.4812 | 0.3498 |
| **AnyUp** | 59.63 | 79.75 | **62.25** | **81.41** | **62.07** | **81.26** | 0.4746 | 0.3364 | **0.4441** | **0.3079** | **0.4455** | **0.3073** |

**Feature Space Preservation.** Ideally, the upsampled features should stay in the same space and distribution as the original, low-resolution features. If that is the case, a linear probe trained on the original feature space should directly transfer to the high-resolution features outputted by a given upsampler for the same input feature extractor, without any finetuning. We test how much different feature upsampling methods preserve the feature distribution, by using linear probes pre-trained on the original DINOv2 (ViT-S) feature extractor (Oquab et al., 2023)[2]. In Tab. 5, we evaluate this for two different tasks, depth estimation on the NYUv2 dataset (Silberman et al., 2012) and semantic segmentation on the ADE20k dataset (Zhou et al., 2018). Qualitative results are in App. E.

Table 5: **Feature Space Preservation.** Semantic Segmentation (ADE20k) and Depth Estimation (NYUv2) with linear probes *pre-trained* on low-resolution DINOv2 features. AnyUp retains the input feature distribution while improving upsampling quality. LoftUp does not retain the input feature distribution, hence its results are heavily degraded. Guided Filtering requires tuning of hyperparameters per sample.

| | Semantic Segmentation | | Depth Estimation | |
| | mIoU (↑) | Acc. (↑) | RMSE (↓) | $\delta_1$ (↑) |
|---|---|---|---|---|
| Bilinear | 39.73 | 73.32 | 0.506 | 0.816 |
| Guided Filter | 37.54 | 72.25 | 0.518 | 0.813 |
| FeatUp | 40.19 | 74.05 | 0.504 | 0.818 |
| LoftUp | 4.27 | 46.58 | 0.765 | 0.802 |
| JAFAR | 39.06 | 73.75 | 0.503 | 0.815 |
| **AnyUp** | **40.83** | **74.94** | **0.498** | **0.822** |

AnyUp preserves the input features the best, while improving the prediction quality at higher resolution through improved upsampling, when compared to learnable and heuristic methods. On the other end of the spectrum, predictions using LoftUp are heavily degraded, which can be explained by the affinity matrix loss employed in its training, aligning intra-image feature similarities instead of directly supervising the predictions with the target features (Huang et al., 2025, Sec. 4.2). We include the implicit feature upsampling version of FeatUp (Fu et al., 2024), whose computational cost also prohibits upsampler-specific probe training, in a similar experiment on images with lower resolution in App. B.2.

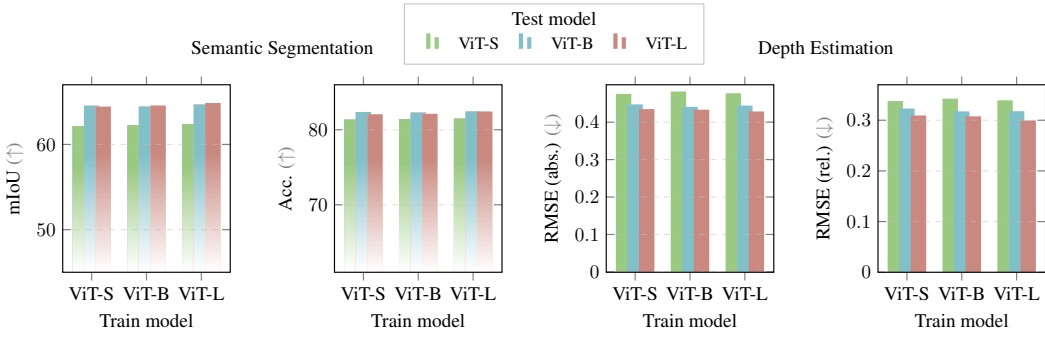

Figure 6: **Generalization to other model sizes.** We vary the model used during upsampler training and the model used for linear probing (test model) for AnyUp. We observe: (1) the general trend ViT-L ≥ ViT-B ≥ ViT-S in linear probing holds no matter the model used in training and (2) there is no significant degradation in upsampling quality when training on a smaller, less powerful ViT.

---

[2]We note that pre-trained linear probes and other feature upsampling methods were not released for the more recent DINOv3 (Xue et al., 2025) method, hence we stick to DINOv2.

**Computational Performance.** The proposed window-based attention used for the main upsampling operation can be implemented efficiently, reducing the runtime and memory requirements by more than 50% compared to prior works JAFAR and LoftUp, while being slightly less efficient than FeatUp. More details on parameter count, FLOPs and runtime, as well as memory usage in forward and backward passes can be found in App. B.1.

## 5.2 FEATURE-AGNOSTIC UPSAMPLING

In Fig. 1, we visualize PCA projections of features, where the *same* feature upsampling model instance, *i.e.*, a model that was trained only on DINOv2 ViT-S features, is applied on features extracted from image encoders ranging from ResNet (He et al., 2016) to DINOv3 (Xue et al., 2025). In Tab. 6, we showcase the remarkable generalization capabilities of AnyUp, where a model trained on DINOv2 and tested on SigLIP 2

Table 6: **Generalization to other models.** AnyUp matches or surpasses the performance of prior upsampling methods while being trained on a fundamentally different feature extractor. When trained with the same feature extractor as used during test time, further small improvements are possible.

|  | Train Model | Test Model | mIoU (Acc) | RMSE (abs / rel) |
|---|---|---|---|---|
| AnyUp | DINOv2 (ViT-S) | SigLIP 2$^{\text{LoftUp}}$ | 51.68 (73.35) | 0.59 / 0.48 |
| AnyUp | SigLIP 2$^{\text{LoftUp}}$ | SigLIP 2$^{\text{LoftUp}}$ | **54.45 (75.49)** | **0.57 / 0.46** |
| LoftUp | SigLIP 2$^{\text{LoftUp}}$ | SigLIP 2$^{\text{LoftUp}}$ | 40.73 (64.87) | 0.72 / 0.60 |
| AnyUp | DINOv2 (ViT-S) | SigLIP 2$^{\text{JAFAR}}$ | 58.51 (78.36) | 0.91 / 0.58 |
| AnyUp | SigLIP 2$^{\text{JAFAR}}$ | SigLIP 2$^{\text{JAFAR}}$ | **60.32 (79.57)** | **0.90 / 0.57** |
| JAFAR | SigLIP 2$^{\text{JAFAR}}$ | SigLIP 2$^{\text{JAFAR}}$ | 60.10 (79.40) | 0.93 / 0.60 |
| AnyUp | DINOv2 (ViT-S) | DINOv3 (ViT-S$^+$) | 62.96 (81.82) | 0.51 / 0.37 |
| AnyUp | DINOv3 (ViT-S$^+$) | DINOv3 (ViT-S$^+$) | 62.99 (81.84) | 0.51 / 0.37 |

outperforms or comes close to the performance of models directly trained on SigLIP 2. Also, a DINOv2-trained AnyUp model generalizes well to DINOv3 features. Finally, as detailed in Fig. 6, AnyUp generalizes well across different DINOv2 architectures.

**Training on Multiple Feature Extractors.** Our feature-agnostic upsampler architecture allows for training on multiple vision backbones. We make use of this property in an experiment to test whether such training can increase generalization performance to completely unseen features at test time. In this experiment, we train AnyUp on features from the following models: DINOv2 (ViT-S), CLIP (ViT-B), SigLIP (ViT-B), DINOv2 w/ registers (ViT-S), and ViT-B trained on ImageNet classification. In Tab. 7, we demonstrate that such training strategy can further increase generalization to unseen test features (*i.e.*, DeiT (ViT-B)), increasing overall performance across test feature extractors. We note that this strategy comes at the price of a small degradation of quality on DINOv2 features when compared against the encoder-specific upsampler, which we attribute to overfitting on the encoder-specific positional encoding artifacts of the latter.

Table 7: **Multi-Backbone Training.** Training on multiple feature extractors achieves comparable performance on seen backbones while improving generalization to unseen ones, outperforming encoder-specific training in cross-backbone evaluation.

| AnyUp trained on: | DINOv2 mIoU (↑) | Acc (↑) | RMSE (abs) (↓) | RMSE (rel) (↓) | SigLIP mIoU (↑) | Acc (↑) | RMSE (abs) (↓) | RMSE (rel) (↓) | DeiT mIoU (↑) | Acc (↑) | RMSE (abs) (↓) | RMSE (rel) (↓) | Average mIoU (↑) | Acc (↑) | RMSE (abs) (↓) | RMSE (rel) (↓) |
|---|---|---|---|---|---|---|---|---|---|---|---|---|---|---|---|---|
| DINOv2 | **62.16** | **81.37** | **0.4755** | **0.3378** | 58.51 | 78.36 | 0.9111 | 0.5780 | 53.93 | 75.90 | 0.6467 | 0.497 | 58.20 | 78.54 | 0.6778 | 0.4709 |
| Multi Backbone | 62.04 | 81.30 | 0.4767 | 0.3392 | **59.52** | **79.07** | **0.9088** | **0.5776** | **54.83** | **76.56** | **0.6418** | **0.4886** | **58.80** | **78.98** | **0.6758** | **0.4685** |

## 5.3 ABLATION STUDY

In Tab. 8, we find that the performance of AnyUp surpasses all its ablations. It is also evidenced that all proposed components lead to notable impact. Note that for the ablation of the data sampling, we replace ours by the simpler training approach of JAFAR (Couairon et al., 2025)).

Remarkably, we observe that when removing the information flow from the input

Table 8: **Ablations.** Effects of removing specific model or training components.

|  | mIoU (Acc.) (↑) | RMSE (abs / rel) (↓) |
|---|---|---|
| **AnyUp** | **62.16 (81.37)** | **0.4755** / 0.3378 |
| w/o window attn. (4.2) | 62.12 (81.34) | 0.4854 / 0.3449 |
| w/o our data sampling (4.3.1) | 62.03 (81.28) | 0.4773 / 0.3387 |
| w/o $L_{\text{self-consistency}}$ (4.3.2) | 62.09 (81.33) | 0.4763 / **0.3363** |
| w/o any regularization (4.3.2) | 61.90 (81.23) | 0.4786 / 0.3401 |
| w/o feature path for key computation | 61.97 (81.23) | 0.4791 / 0.3441 |

feature map to the key computation in the upsampling model (see Fig. 3), we are still able to obtain results that match the performance of prior feature upsampling methods, see, *e.g.*, RMSE for absolute depth estimation in Tab. 3. This indicates that an upsampling strategy based only on position and color matching between a high-resolution image and its downsampled equivalent can already provide a relatively strong prior for feature upsampling when trained with our pipeline.

To further explore this property and quantify the effectiveness of our proposed feature-agnostic layer (Sec. 4.1), we vary the numbers of basis filters in an additional experiment. As can be seen in Tab. 9, upsampling performance increases with larger filter basis size in the feature-agnostic layer, thus validating our hypothesis that a successful upsampler also needs to be able to see the low resolution feature map to distinguish between image areas that are of similar color but of different semantics. We decide to use a filter basis size of $M = 128$ in our implementation.

Table 9: **Impact of Filter Basis Size on Performance.** Scores highlighted from best (gray) to worst (white). We show the previous state-of-the-art results in a separated line for comparison.

| $M$ | mIoU ($\uparrow$) | Acc ($\uparrow$) | RMSE (abs) ($\downarrow$) | RMSE (rel) ($\downarrow$) |
|---|---|---|---|---|
| 256 | 62.19 | 81.38 | 0.4765 | 0.3371 |
| **128** | 62.16 | 81.37 | 0.4755 | 0.3378 |
| 64 | 62.09 | 81.34 | 0.4759 | 0.3380 |
| 32 | 62.07 | 81.31 | 0.4779 | 0.3402 |
| 16 | 62.09 | 81.32 | 0.4773 | 0.3404 |
| 8 | 62.00 | 81.27 | 0.4792 | 0.3415 |
| 0 | 61.97 | 81.23 | 0.4791 | 0.3441 |
| SOTA | 62.15 | 81.32 | 0.4816 | 0.3413 |

## 6 LIMITATIONS

FeatSharp (Ranzinger et al., 2025) combines feature upsampling with explicitly de-noising the input features from positional encoding artifacts (Yang et al., 2024). We note that this learned per-model denoising step on the input features can be easily prepended to our pipeline but as the learned denoising weights have not been made public at the time of submission, we do not include this analysis in our experiments. Further, through our training strategy on image parts, our model is effectively also trained to mostly ignore such positional encoding artifacts.

In addition, our upsampling approach based on a single image-to-features-attention is computationally efficient and outperforms previous works while being more general. However, it relies on the simplifying assumption that upsampled features can be computed as linear combinations of low-resolution input features: A patch feature almost certainly encodes sub-patch-level spatial information in its high-dimensional channels, see, *e.g.*, works trying to reconstruct RGB images from patch features (Bordes et al., 2022,). Such information could likely be extracted and used in feature upsampling when using a larger and more complex upsampling model.

## 7 CONCLUSION

We introduced AnyUp, a first-of-its-kind method for feature upsampling from any resolution to any resolution, which generalizes to feature representations that it was not trained on. Key technical novelties include a feature-agnostic layer, windowed attention, and a training strategy, which work together to achieve state-of-the-art upsampling quality. We make our code and models publicly available at https://github.com/wimmerth/anyup.

**Acknowledgements** This work was partially funded by the Saarbrücken Research Center for Visual Computing, Interaction, and Artificial Intelligence (VIA). Thomas Wimmer is supported by the Max Planck ETH Center for Learning Systems. Jan Eric Lenssen is supported by the German Research Foundation (DFG) - 556415750 (Emmy Noether Programme, project: Spatial Modeling and Reasoning).

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

# A  ADDITIONAL DETAILS ON RELATED WORK

**FeatUp** (Fu et al., 2024): Fu et al. (2024) propose training the upsampler in a "multi-view reconstruction" setting, where the feature computation and learned upsampling should be equivariant to small image augmentations, using a learned feature downsampling operation. FeatUp consists of multiple stacked joint bilateral upsampling (JBU) layers making upsampling only possible for integer factors.

**LoftUp** (Huang et al., 2025): Huang et al. (2025) propose a stacked high-res to low-res attention. Their training strategy consists of two stages: First, SAM (Kirillov et al., 2023) is used for object mask generation. Low-resolution features are then blended per-mask using bicubic upsampling. In their second stage, self-distillation with student and teacher models is performed. Instead of using a cosine-similarity loss to compare their upsampled predictions with the ground truth, Huang et al. (2025) propose an affinity matrix loss, where the intra-image pixel feature similarity matrices are matched. This loss is computed on high-resolution images.

**JAFAR** (Couairon et al., 2025): Couairon et al. (2025) propose feature upsampling as a single high-res to low-res attention. Training is performed on low resolutions and small scale changes (up to 4x). While a spatial semantic feature modulation (Wang et al., 2018) is used to merge information from the downsampled image and the input feature map in JAFAR, we find that this is not giving noticeable improvements and replace it with a simple concatenation of features followed by a standard ResNet block.

**FeatSharp** (Ranzinger et al., 2025): FeatSharp builds upon FeatUp (Fu et al., 2024), where tiling of the input image and multiple forward passes of the encoder model are used to get higher-resolution outputs. De-biasing from positional encoding artifacts is learned. The weights for the upsampler are not openly available, hence we do not compare against this method.

# B  ADDITIONAL EXPERIMENTS

## B.1  PERFORMANCE ANALYSIS

We summarize performance in terms of runtime and peak memory usage in Tab. 10 (measured on a single NVIDIA A100 GPU). Global attention requires $H \times W \times h \times w$ computations, while window attention, using windows sized $\sigma$ (relative to the input size), significantly reduces the requirement to only $H \times W \times \sigma h \times \sigma w$ computations. In practice, we base our implementation on the $\mathcal{N}$ATTEN library (Hassani et al., 2025,), which implements CUDA kernels for several architectures that make use of the sparsity in window attention and set $\sigma = 0.2$. We further extend our code to enable trade-off between memory and speed through chunked computation of the attention that enables even larger input images and feature maps.

Table 10: **Performance Analysis.** AnyUp benefits from the sparsity of the window attention.

(a) Parameter count and inference time (ms) / FLOPs comparison (GFLOPs) by resolution.

| Model | # Params (M) | $112^2$ Time (ms) | $112^2$ FLOPs | $224^2$ Time (ms) | $224^2$ FLOPs | $448^2$ Time (ms) | $448^2$ FLOPs |
|---|---|---|---|---|---|---|---|
| FeatUp | 0.2 | 8.6 | 2.6 | 20.6 | 10.4 | 73.7 | 41.6 |
| LoftUp | 4.3 | 8.2 | 47.1 | 33.5 | 204.1 | 201.2 | 1065.3 |
| JAFAR | 0.7 | 4.3 | 5.5 | 19.6 | 26.8 | 113.9 | 186.1 |
| **AnyUp** | 0.8 | 2.7 | 5.2 | 12.8 | 20.6 | 92.4 | 82.4 |

(b) Memory usage (GB) by resolution for forward / backward passes.

| Model | $112^2$ Fwd | $112^2$ Bwd | $224^2$ Fwd | $224^2$ Bwd | $448^2$ Fwd | $448^2$ Bwd |
|---|---|---|---|---|---|---|
| FeatUp | 0.3 | 0.4 | 0.9 | 1.4 | 3.4 | 5.6 |
| LoftUp | 0.2 | 0.7 | 0.8 | 2.8 | 7.9 | 21.1 |
| JAFAR | 0.2 | 0.5 | 0.6 | 2.7 | 6.9 | 22.3 |
| **AnyUp** | 0.2 | 0.8 | 0.8 | 3.3 | 3.3 | 12.9 |

## B.2  COMPARISON AGAINST FEATUP'S IMPLICIT UPSAMPLING

Due to the computational burden of FeatUp's proposed implicit upsampling, where high-resolution features are optimized in a NeRF-like manner at inference time, it is infeasible to train linear probes for semantic segmentation or depth/normal estimation, as done in Sec. 5.1. Instead, we analyze its performance when used with pre-trained linear probes from DINOv2, as in Tab. 5. We show the results in Tab. 11. Note, that we evaluate on images of 224px instead of 448px used in Tab. 5, as

Table 11: **Comparison against FeatUp (implicit).** Feature space preservation comparison with pre-trained linear probes for DINOv2 ViT-S features when upsampling to 224px.

|  | Semantic Segmentation | | Depth Estimation | |
|---|---|---|---|---|
|  | mIoU ($\uparrow$) | Acc. ($\uparrow$) | RMSE ($\downarrow$) | $\delta_1$ ($\uparrow$) |
| Bilinear | 35.54 | 70.20 | 0.510 | 0.818 |
| Guided Filter | 32.17 | 68.25 | 0.537 | 0.802 |
| FeatUp | 35.60 | 71.40 | 0.510 | 0.816 |
| $\rightarrow$ implicit | 36.70 | 70.95 | 0.513 | 0.818 |
| LoftUp | 3.82 | 44.35 | 0.534 | 0.803 |
| JAFAR | 37.52 | 72.64 | 0.505 | **0.822** |
| **AnyUp** | **37.91** | **72.78** | **0.502** | **0.822** |

the inference-time optimization of FeatUp is too costly to run on higher resolutions. As in Tab. 5, AnyUp outperforms all prior works when using pre-trained probes.

## C  IMPLEMENTATION DETAILS

We train our method on the ImageNet dataset (Krizhevsky et al., 2017) for a total of 100,000 training steps. We choose a batch size of 4 with 4 random local crops per training image. Training takes around 5 hours on a single NVIDIA-H100 GPU. We use the AdamW optimizer (Loshchilov & Hutter, 2019) with a learning rate of 2e-4 and a batch size of 4.

We note that the results for semantic segmentation linear probing reported in Sec. 5.1 deviate from the results reported by Couairon et al. (2025). This is caused by fixing a bug in the probing training compared to their implementation. We ran all experiments with the official published weights for concurrent works.

### C.1  SELF-CONSISTENCY REGULARIZATION AND DATA AUGMENTATIONS

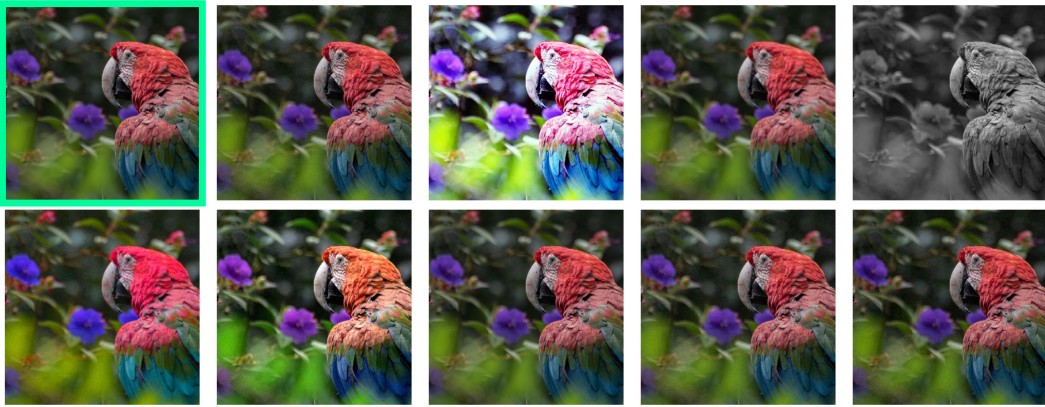

Figure 7: Randomly sampled augmentations applied on a test image (top left).

We add a self-consistency regularization $L_{\text{self-consistency}}$, also proposed in prior work (Couairon et al., 2025), where we increase the level and diversity of data augmentations to increase the robustness of our upsampling method. We show a batch of randomly sampled augmented images in Fig. 7. The self-consistency regularization is computed as

$$L_{\text{self-consistency}} = d_{\text{cos-mse}} \left( f(p, I_{hr}), f(p, I'_{hr}) \right), \tag{3}$$

where $I'_{hr}$) is the noised/augmented version of the high-resolution image. The self-consistency regularization does not rely on ground-truth features $\hat{q}$ and is thus computed at higher resolution, *i.e.*, 224x224.

## D  COMPULSORY NOTE ON LLM USAGE

No large language models were used in writing or ideation of this paper except as aid for styling Fig. 6.

## E  ADDITIONAL VISUALIZATIONS

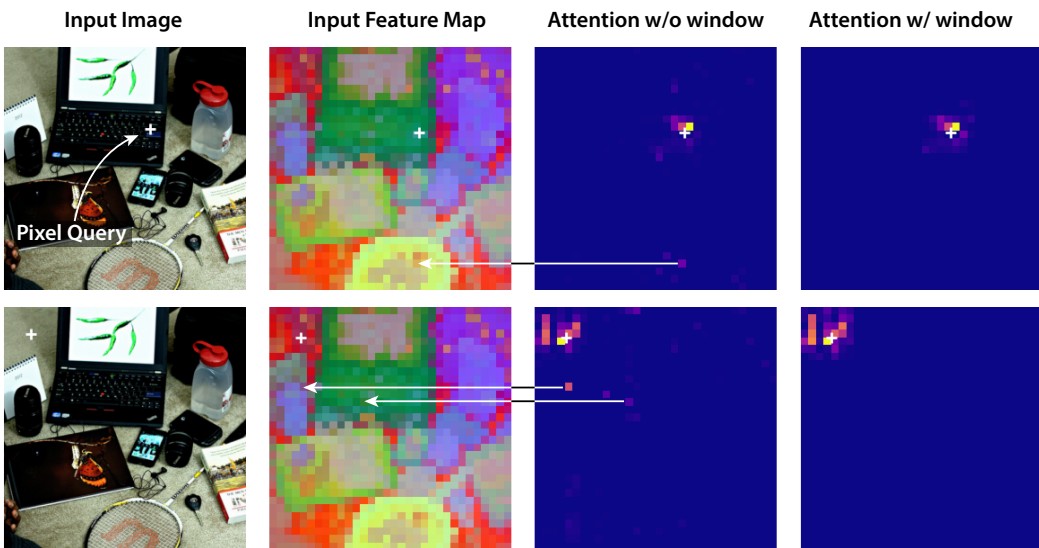

Figure 8: Visualization of attention artifacts and removal thereof through local window attention. Unconstrained, global attention leads to upsampled features relying on information from far-away, non-related objects. We provide a simple fix to this which also simplifies the upsampling problem for the model by restricting attention only to local windows that are computed relative to the feature map size.

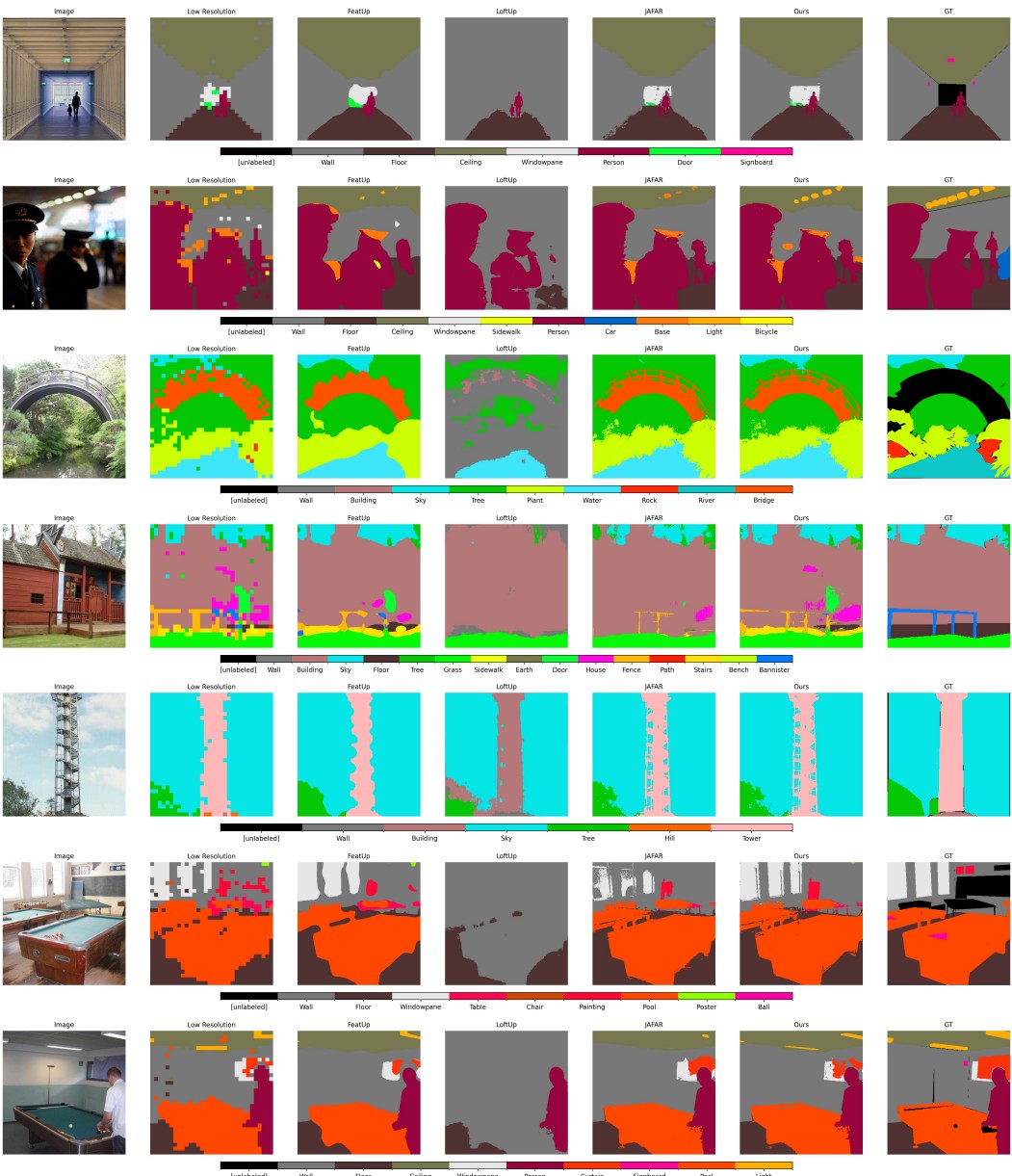

Figure 9: Linear Probing results for semantic segmentation using a pre-trained DINOv2 (ViT-S) probe. AnyUp preserves the input feature space while upsampling and sometimes even outperforming the ground-truth, *e.g.*, the tower segmentation in the fifth row. While FeatUp's predictions are generally consistent with the low-resolution predictions, it smoothens object boundaries too much and is inable to get sharp segmentations. LoftUp shifts the feature distribution while upsampling, resulting in erroneous predictions with the pre-trained probe. Predictions using JAFAR suffer from reduced details for thin objects, see, *e.g.*, the lights in row 2 or the fence in row 4.

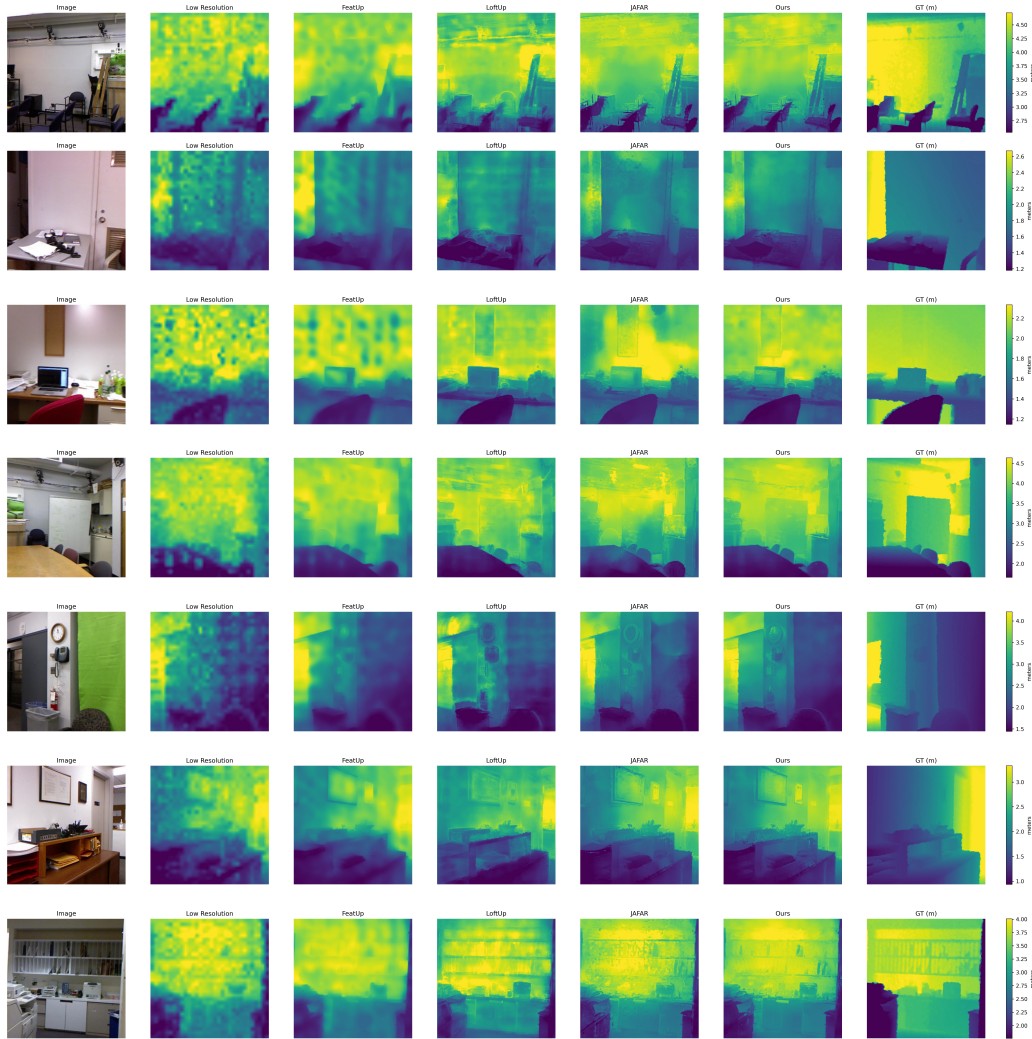

Figure 10: Linear Probing results for depth estimation using a pre-trained DINOv2 (ViT-S) probe. AnyUp stands out by preserving sharp edges from the guidance image while preserving the locality of features needed for smooth depth map prediction and preserving full objects, as, *e.g.*, the white board in the fourth row. Note that the feature distribution shift of LoftUp is less pronounced for depth estimation, which can be partly attributed to the scale-shift alignment performed after prediction.

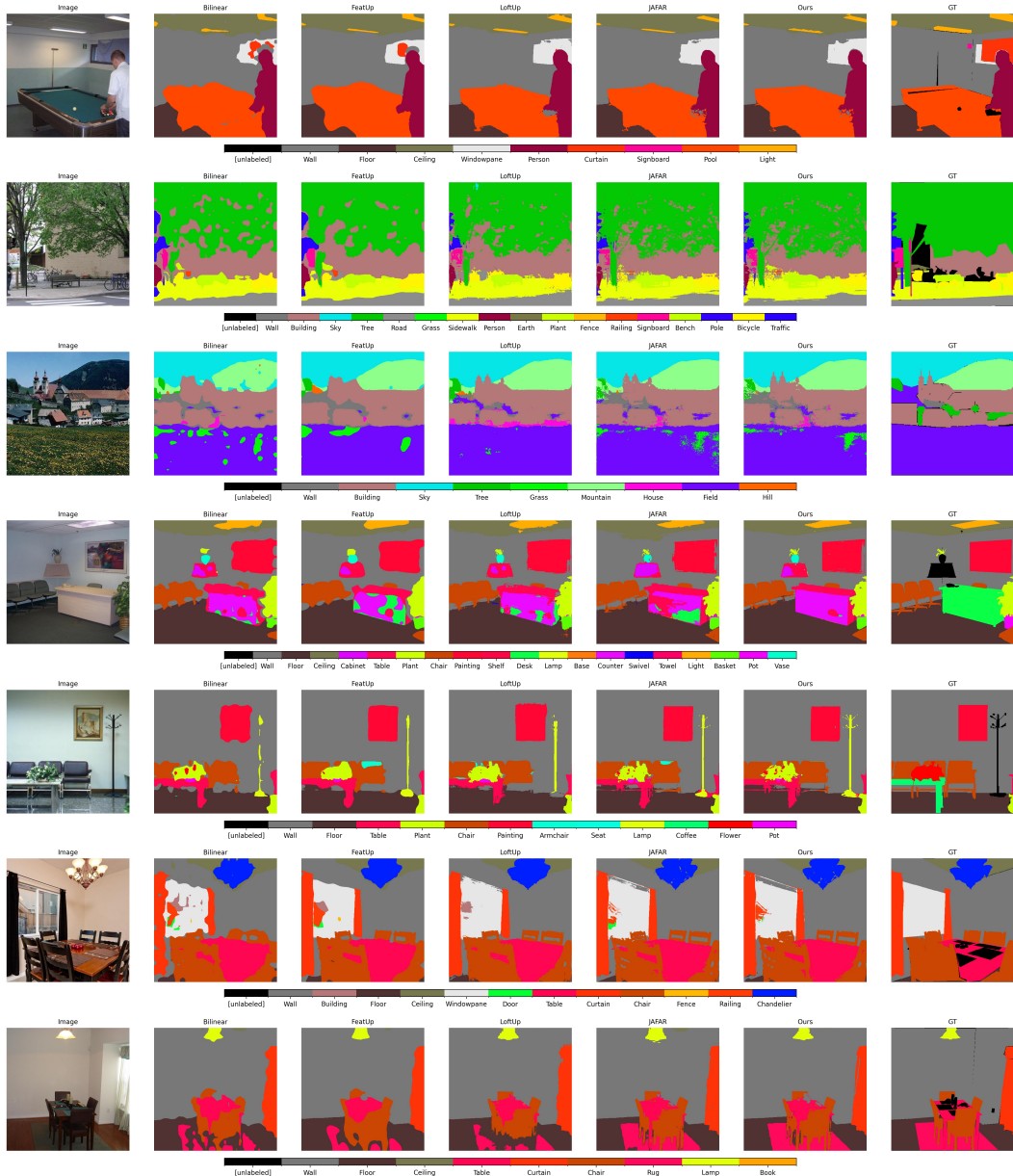

Figure 11: Additional linear probing results for semantic segmentation using probes trained on upsampled DINOv2 (ViT-S) features. AnyUp outperforms prior upsampling methods by also being able to segment fine-details and giving cleaner segmentation outputs.

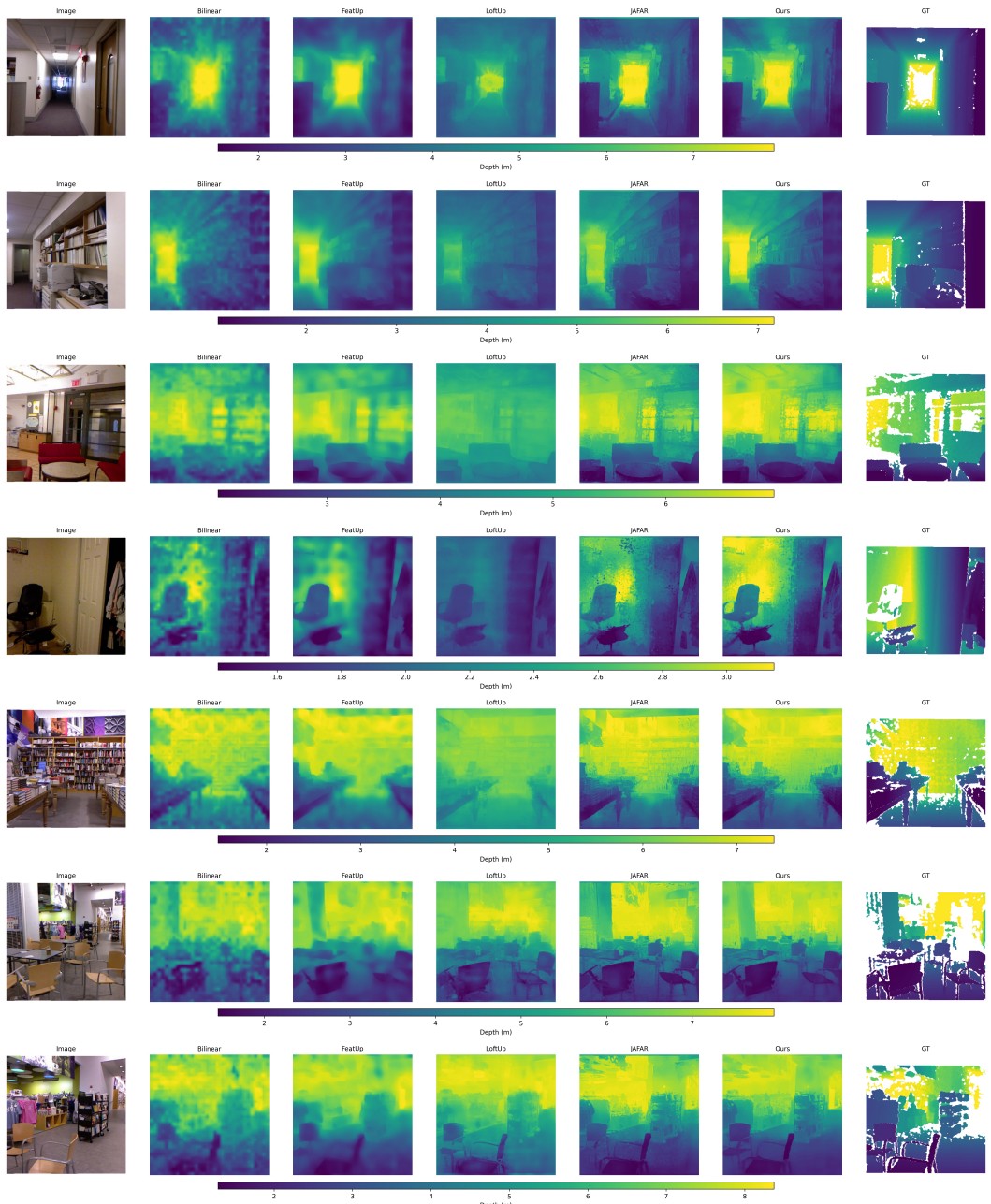

Figure 12: Additional linear probing results for depth estimation using probes trained on upsampled DINOv2 (ViT-S) features. AnyUp consistently gives sharp object boundaries while matching the ground truth the best.

