# OpenReview forum: "AnyUp: Universal Feature Upsampling"
_ICLR.cc/2026/Conference — ICLR 2026 Oral_

### Official Review · Reviewer_spdT · 2025-10-29

**Soundness:** 3
**Presentation:** 2
**Contribution:** 4
**Rating:** 6
**Confidence:** 3

**Summary:**

This paper proposes a feature upsampling method applicable to any visual feature at any resolution without encoder-specific training. The Feature-Agnostic Layer achieves applicability to feature dimensions. The results show that it achieves state-of-the-art performance on multiple datasets.

**Strengths:**

1. This method makes a great contribution to the community. It solves the problem that previous methods require re-training for specific encoders and achieves more universal upsampling.
2. The article is well-written and easy to understand.
3. Feature-Agnostic Layer is a novel, concise and effective module.
4. It can be seen from the experimental results that AnyUp has achieved SOTA performance.

**Weaknesses:**

Some details of the methods were not explained clearly. Please refer to the section Questions.

**Questions:**

1. This LOCAL WINDOW ATTENTION cannot capture long-range dependencies like GLOBAL ATTENTION can. Will this lead to performance degradation?
2. Are these artifacts introduced by the GLOBAL ATTENTION itself? Please analyze the reasons.
3. What is the significance of sampling the image and then supervising the local part? Is it merely for being more lightweight and efficient?
4. Table 7 shows that the model with data sampling has better performance. Logically, the unsampled supervision information should be more comprehensive and should yield better results. Could you explain why data sampling has better performance?

---

> ### Author Response · Authors · 2025-11-24
>
> We thank the reviewer for their positive feedback, stating that AnyUp is a **great contribution to the community**, the **paper is well-written** and that the proposed feature-agnostic layer is **novel, concise and effective**.
>
> > **Q1: Does the local window attention lead to performance degradation?**
>
> As can be seen in Tab. 7 of the paper, the opposite is true: Using the proposed window attention instead of global attention for upsampling results in improved performance on downstream tasks.
>
> > **Q2: Are these artifacts introduced by the global attention itself?**
>
> We thank the reviewer for this question and would like to emphasize that the output of the attention is the final upsampled feature output.
> Therefore, the answer is yes, the artifacts shown in Fig. 8 are introduced by the fact that the attention is global.
> Interpreting the depicted attention maps is easier than for normal transformers, as the relative amount of attention that one token is paying to another is directly linked to the factor of how much this token will contribute to the upsampled feature at the query location using the weighted sum.
> Our proposed local window attention restricts attention to local neighborhoods, thus effectively prohibiting any contributions from far-away image areas.
>
> > **Q3 & Q4: What is the significance of the proposed data sampling and training supervision using local image crops? Why does it perform better than a model without this training strategy?**
>
> Great question! We provide two explanations for this behavior:
>
> First, our model implicitly learns to denoise artifact tokens stemming from ViT positional encodings and attention sinks. This is generally beneficial for better performance on dense tasks [1].
> Why is that? Prior works found that such artifacts always occur at the same relative position in the images [1,2,3]. When extracting features on a local image crop, the relative pixel positions in the source (global) image do not match the target (local) image anymore. Therefore, the upsampling model learns to be robust against this “noise”.
>
> Secondly, most ViTs, like the DINOv2 model which was used as a feature extractor during training in most of the experiments, are trained on a fixed image resolution (224x224).
> Increasing the resolution to compute the target features for upsampler training (as done in the simple training strategy) can result in undesired feature quality degradation, especially in the case of using learned positional encodings as in DINOv2. In contrast, our crop-based training strategy successfully avoids this by only querying the feature backbone at its (lower) training resolution.
>
> [1] Yang, Jiawei, et al. "Denoising vision transformers." European Conference on Computer Vision. Cham: Springer Nature Switzerland, 2024.
>
> [2] Jiang, Nick, et al. "Vision Transformers Don't Need Trained Registers." Advances in Neural Information Processing Systems, 2025.
>
> [3] Ranzinger, Mike, et al. "FeatSharp: Your Vision Model Features, Sharper." Forty-second International Conference on Machine Learning, 2025.

---

> ### Comment · Reviewer_spdT · 2025-11-28
>
> Thanks for your active rebuttal and for answering my questions. Your work is of great value to our community.

---

### Official Review · Reviewer_9SrJ · 2025-10-30

**Soundness:** 3
**Presentation:** 3
**Contribution:** 3
**Rating:** 6
**Confidence:** 4

**Summary:**

This paper presents AnyUp, a universal upsampling framework that can upsample features from any encoder to any resolution. This paper contributes i) a feature-agnostic convolutional layer independent of feature dimensions, ii) a local window attention mechanism to avoid irrelevant matches introduced by global attention in space, improving the quality and efficiency of upsampling, and iii) a lightweight training strategy based on local image cropping to get rid of generating high-resolution features as supervision. Experimental results show that AnyUp achieves state-of-the-art performance on several segmentation, surface normal estimation and depth estimation benchmarks. Crucially, AnyUp can generalize effectively to vision encoders unseen during training.

**Strengths:**

1.  Clear Motivation and Significant Value: The "encoder-specific" problem is a real pain point when applying recent model-agnostic upsamplers. A "train once, use anywhere" upsampler is of high practical value for the dense prediction community.
2.  Comprehensive Experimental Validation: The paper conducts extensive comparisons against strong baselines on multiple benchmarks (COCO, PASCAL-VOC, ADE20k, NYUv2), consistently achieving SOTA or highly competitive performance. Generalization experiments also demonstrate the model’s transferable ability on different feature extractors.
3.  Insightful Analysis of Feature Space Preservation: The feature space preservation experiment in Table 5 is an interesting supplementary analysis. Beyond performance gains, it reveals why AnyUp succeeds—by maximally preserving the semantic distribution of the original features during upsampling, which offers more insights than simple metrics.

**Weaknesses:**

1.  Insufficient Analysis of the Feature-Agnostic Layer: How sensitive is the model to the hyperparameter M (the number of basis filters)? What kind of visual patterns do the learned basis filters (ψj) actually capture? How much performance improvement can feature-agnostic layers itself bring? What is the performance difference between the proposed feature-agnostic layer and the standard convolution? Experiments on these factors are expected for a comprehensive understanding.
2.  Parameters and inference time should be analyzed. As an inherited version from JAFAR, the number of parameters of AnyUp and the inference time should be discussed, just like JAFAR did.
3.  A key design choice is that the Value vector (V) in the attention mechanism directly uses "unprocessed input features" (line 197), justified as a way to preserve the feature space. This assertion lacks direct experimental evidence (e.g. What impact will inserting a convolution layer in the value-branch could have on performance?).
4.  Ambiguous terms used in the motivation table: what is the meaning of “any task”? If I interpret correctly, most upsamplers (e.g., Dysample) can be used for different tasks. It is just a matter of improved or decreased performance. It seems that the paper would like to say whether an upsampler should be retrained on a new task. This is not what ‘any task’ means.
5.	To strengthen the comparison of the encoder-agnostic property of AnyUp in Table 6, it is better to include results from other upsampling models when they operate on features from different encoders. Specifically, these models could be generalized to different encoders in a training-free manner (e.g., via channel-wise linear interpolation) for a supplementary comparison.
6.	Some typographical and formatting errors: line-090, “it is was not trained on”, line-242, “the query point (Ramachandran et al., 2019)., As a high” (This point is not considered for rating)

**Questions:**

As the paper's most central technical novelty, the Feature-Agnostic Layer (Section 4.1) is clearly explained in principle, but lacks a deep experimental analysis of its internal mechanism.

---

> ### Author Response · Authors · 2025-11-24
>
> We thank the reviewer for the valuable feedback and the positive review, acknowledging the **clear motivation and significant value of AnyUp**, as well as the **SOTA performance** and **insightful analysis on feature space preservation**.
>
> > **W1 & Q1: Further Analysis on the feature-agnostic layer.**
>
> As can be seen in Tab. 7 of the main paper, an upsampler (using window attention to upsample and trained using our proposed training scheme) that only “sees” the high-resolution guidance image and its downsampled equivalent ($M=0$) is already a quite capable system that is able to upsample features fairly well. However, when not using our feature-agnostic layer, its performance still lags behind the previous state of the art backbone-specific upsampling methods. We hypothesize that this is caused by the fact that a successful upsampler also needs to be able to see the low resolution feature map to distinguish between image areas that are of similar color but of different semantics. When increasing the number of basis filters in our feature-agnostic layer (see Table below), we observe that the upsampling performance increases, which validates our hypothesis. While the learned patterns in the filter basis are not directly interpretable qualitatively, AnyUp using the feature-agnostic layer quantitatively outperforms a “feature-blind” comparable version across all tasks.
>
> **Table 1: Impact of filter basis size on performance.**
>
> |      | Semantic Segmentation |                  |      Depth Estimation     |                           |
> |:----:|:---------------------:|:----------------:|:-------------------------:|:-------------------------:|
> |   **M**  |   **mIoU** ($\uparrow$)   | **Acc** ($\uparrow$) | **RMSE (abs)** ($\downarrow$) | **RMSE (rel)** ($\downarrow$) |
> |  256 |         **62.19**     |     **81.38**    |           0.4765          |         **0.3371**        |
> |  128 |     $\underline{62.16}$ | $\underline{81.37}$ |       **0.4755**      |     $\underline{0.3378}$  |
> |  64  |         62.09         |       81.34      |   $\underline{0.4759}$    |           0.3380          |
> |  32  |         62.07         |       81.31      |           0.4779          |           0.3402          |
> |  16  |         62.09         |       81.32      |           0.4773          |           0.3404          |
> |   8  |         62.00         |       81.27      |           0.4792          |           0.3415          |
> |   0  |         61.97         |       81.23      |           0.4791          |           0.3441          |
> | SOTA |         62.15         |       81.32      |           0.4816          |           0.3413          |
>
> We would like to further clarify that the requested ablation - using a standard convolution instead of our feature-agnostic layer - is not possible, as a standard convolution is not agnostic to the number of input channels in the input, a key objective of our proposed feature-agnostic layer.

---

> ### Author Response · Authors · 2025-11-24
>
> > **W2: Parameter count and inference time analysis.**
>
> We summarize performance in terms of runtime, number of parameters and peak memory usage in the table below (measured on a single NVIDIA A100 GPU).
> Global attention requires $H \times W \times h \times w$ computations, while window attention, using windows sized $\sigma$ (relative to the input size), significantly reduces the requirement to only $H \times W \times \sigma h \times \sigma w$ computations, which is reflected in both the computational runtime, as well as the peak memory usage.
> In practice, we base our implementation on the NATTEN library [1], which implements CUDA kernels for several architectures that make use of the sparsity in window attention and set $\sigma=0.2$.
> We further extend our code to enable trade-off between memory and speed through chunked computation of the attention that enables even larger input images and feature maps (not used in the benchmarking below).
> As can be seen in Tab. 7 in the main paper, using the proposed window attention instead of global attention for upsampling results in improved performance on downstream tasks.
>
> [1] Hassani, Ali, et al. "Generalized Neighborhood Attention: Multi-dimensional Sparse Attention at the Speed of Light." arXiv preprint arXiv:2504.16922 (2025).
>
> **Table 2: Parameter count and inference time / FLOPs comparison by resolution**
>
> |            | **Resolution**   | **112²**      | **112²**   | **224²**      | **224²**   | **448²**      | **448²**   |
> |------------|------------------|---------------|------------|---------------|------------|---------------|------------|
> | **Model**  | **# Params (M)** | **Time (ms)** | **GFLOPs** | **Time (ms)** | **GFLOPs** | **Time (ms)** | **GFLOPs** |
> | **FeatUp** | 0.2          | 8.6           | **2.6**    | 20.6          | **10.4**   | **73.7**      | **41.6**   |
> | **LoftUp** | 4.3              | 8.2           | 47.1       | 33.5          | 204.1      | 201.2         | 1065.3     |
> | **JAFAR**  | 0.7| $\underline{4.3}$ | 5.5    | $\underline{19.6}$ | 26.8   | 113.9         | 186.1      |
> | **AnyUp (Ours)**   | 0.8      | **2.7**       | $\underline{5.2}$ | **12.8**  | $\underline{20.6}$ | $\underline{92.4}$ | $\underline{82.4}$ |
>
> We can see that in terms of runtime and FLOPS, AnyUp is quite efficient and faster than all previous approaches except FeatUp, which is slightly faster on some resolutions.
>
> **Table 3: Memory usage (GB) by resolution for forward / backward passes**
>
> | **Resolution** | **112²**              | **112²**          | **224²**          | **224²**          | **448²**          | **448²**          |
> |----------------|-----------------------|-------------------|-------------------|-------------------|-------------------|-------------------|
> | **Model**      | **Fwd**               | **Bwd**           | **Fwd**           | **Bwd**           | **Fwd**           | **Bwd**           |
> | **FeatUp**     | $\underline{0.3}$     | **0.4**           | 0.9               | **1.4**           | $\underline{3.4}$ | **5.6**           |
> | **LoftUp**     | **0.2**               | 0.7               | $\underline{0.8}$ | 2.8               | 7.9               | 21.1              |
> | **JAFAR**      | **0.2**               | $\underline{0.5}$ | **0.6**           | $\underline{2.7}$ | 6.9               | 22.3              |
> | **AnyUp (Ours)**       | **0.2**       | 0.8               | $\underline{0.8}$ | 3.3               | **3.3**           | $\underline{12.9}$ |
>
> Similarly to runtime, AnyUp's memory consumption is lower than that of LoftUp and JAFAR and slightly higher than FeatUp's for some resolutions.

---

> ### Author Response · Authors · 2025-11-24
>
> > **W3: Impact of using raw features as values. How will inserting a convolutional layer in the value-branch affect performance?**
>
> We motivate the use of raw features as values with the observation that LoftUp, which processes the input features during the upsampling with multiple attention layers, fails to preserve the original feature space, as can be seen in Tab. 5 of the paper.
> In combination with our local window attention, we effectively prevent major feature distribution shifts during upsampling and make our model more reliable in out-of-distribution settings, where potential overfitting on the training feature space could result in worse performance when features are further processed before / during upsampling.
>
> More importantly, using the raw features as values in our upsampling attention allows AnyUp to stay feature-agnostic without any complications. To preserve this property when applying additional processing, we would need a layer or network that operates independently of input dimensionality, is invariant to channel order, and preserves the structure of the original feature space.
>
> > **W4: Ambiguous terminology in the related work section.**
>
> We thank the reviewer for their feedback regarding this.
> We denote methods that are proposed as building blocks to certain specialized models, rather than general feature upsamplers, as not being task-agnostic. This aligns with the interpretation in previous works, see e.g. Sec. 2 in [1] or Sec. 2 in [2].
>
> [1] Couairon, Paul, et al. "JAFAR: Jack up Any Feature at Any Resolution." Advances in Neural Information Processing Systems, 2025.
>
> [2] Huang, Haiwen, et al. "LoftUp: Learning a Coordinate-Based Feature Upsampler for Vision Foundation Models." IEEE/CVF International Conference on Computer Vision, 2025.

---

> ### Author Response · Authors · 2025-11-24
>
> > **W5: Comparison against prior works when using features from different encoders.**
>
> We would like to highlight that AnyUp is the first method that is feature-agnostic and that this is a key difference to prior learnable methods.
> Interpolating the channel dimensions to fit the upsampler’s training features shape naturally results in information loss, which can’t be fully recovered after upsampling.
>
> When taking, for example, SigLIP ViT-B features and interpolating them to a lower feature dimensionality (768 to 384), applying an upsampler model like JAFAR or LoftUp (which was trained on 384 dim. features, e.g., from DINO ViT-S), will result in 384.-dim. output features.
> Mapping back to the input feature space (768-dim.), i.e. through linear upsampling, is not fully possible as the initial interpolation is not invertible without loss of information.
>
> To demonstrate this, we run an experiment where we follow the procedure described above. More specifically, we train linear probes on the upsampled feature maps from FeatUp, LoftUp and JAFAR, where we use the upsampler models trained on DINOv2 ViT-S features and test on SigLIP 2 ViT-B features.
> Our results show that, while some content is preserved in the upsampled features, the linear probing results are significantly worse than with our proposed feature-agnostic AnyUp model that was also trained only on DINOv2 ViT-S features.
>
> **Table 4: Generalization comparison against prior works**
>
> |           |              | SigLIP ViT-B |       |            |            |
> |-----------|--------------|--------------|-------|------------|------------|
> | **Upsampler** | **Train Model**  | **mIoU** ($\uparrow$) | **Acc** ($\uparrow$) | **RMSE (abs)** ($\downarrow$) | **RMSE (rel)** ($\downarrow$) |
> | FeatUp    | DINOv2 ViT-S | 55.71 | 76.72 | 0.9265 | 0.5864 |
> | LoftUp    | DINOv2 ViT-S | 55.38 | 76.56 | 0.9451 | 0.6192 |
> | JAFAR     | DINOv2 ViT-S | 53.55 | 74.85 | 0.9433 | 0.6058 |
> | AnyUp (Ours) | DINOv2 ViT-S | **58.51** | **78.36** | **0.9111** | **0.5780** |
>
> > **W6: Typographical Errors**
>
> We thank the reviewer for catching these typos and will fix them in the updated paper.

---

### Official Review · Reviewer_MPXT · 2025-10-30

**Soundness:** 3
**Presentation:** 3
**Contribution:** 3
**Rating:** 8
**Confidence:** 4

**Summary:**

This work introduces AnyUp, an algorithm for upsampling deep vision network representations while retaining their semantics. Most importantly, AnyUp expands beyond other works in the literature by introducing a layer that once-trained will work on any deep network's features. They achieve this with the introduction of a new windowed guided upsampling layer that operates on features of any dimensionality. They justify their work with a suite of experiments that test the ability of features to be used in downstream dense prediction tasks, the ability to be used as a drop-in replacement for existing tasks, and the ability to generalize across networks and resolutions.

**Strengths:**

- Simple and elegant idea that is of interest to the community
- Good comparisons to prior work and nice panoply of experiments
- Nice ablations and thorough investigations of transfer to other backbones and resultions
- Good writing and presentation

**Weaknesses:**

- Nit: line 53: some prior methods like FeatUp dont require evaluating the backbone on higher resolution images but rather small pixel jitters
- Your explanation of the feature agnostic upsampler is a little hard to understand from the small paragraph provided, consider expanding this to provide a bit more intuition behind the math here. Also see questions section for another possible way to provide intuition behind this core contributionm
- FeatUp's implicit version, which performs better than its JBU version does not seem to be compared against in this work, this method is also allows for any resolution inference, though at a time cost to perform the implicit optimization
- Consider adding flop / peak mem comparisons
- Some prior works evaluate in the joint training context to demonstrate that upsamplers can be used as a replacement for resize convs in unets. One might consider that experimental setting here as well.

**Questions:**

- How does AnyUp handle tokens that are non-spatial like those in DinoV2 before the paper Vision Transformers need registers
- During training AnyUp uses crops of the original image to help it learn to reconstruct the intricacies of high resolution features, to what extent does this reliance on image parts change its semantics (i.e the features for an object can change when that object is small or large relative to the field of view)
- What does AnyUp learn that enables such promising transfer? What do these filters learn that enable them to upsample? Perhaps adding a section in the appendix might help the curious get a sense for what the parameters you add learn.

---

> ### Author Response · Authors · 2025-11-24
>
> We thank the reviewer for the insightful and positive review, acknowledging that **AnyUp is simple, elegant and of interest to the community**, the paper contains good comparisons to prior work, thorough investigations of transfer to other backbones and resolutions and that the paper is well-written.
>
> > **Some prior methods don’t require backbone evaluation at higher resolution during training.**
>
> Thank you for this remark. We will rephrase the corresponding paragraph in the updated paper.
>
> > **Missing Comparison against FeatUp’s implicit version.**
>
> We report a comparison against FeatUp’s implicit version in the table below.
> Due to the computational burden of FeatUp's proposed implicit upsampling, taking about 13 minutes of optimization per image in our setup, where high-resolution features are optimized in a NeRF-like manner at inference time, it is infeasible to train linear probes for semantic segmentation or depth/normal estimation on upsampled features, as done in Sec. 5.1.
> Instead, we analyze its performance when used with pre-trained linear probes from DINOv2, as in Tab. 5.
> Note, that we evaluate all models on images of 224px instead of 448px, as the inference-time optimization of FeatUp is too costly to run on higher resolutions.
> We find that it slightly outperforms or matches the performance of the feed-forward FeatUp version but fails to match AnyUp’s results.
>
> **Table 1: Comparison against FeatUp (implicit) using pre-trained linear probes**
>
> |                   | Semantic Segmentation |       | Depth Estimation |            |
> |-------------------|:---------------------:|:-----:|:----------------:|:----------:|
> |                   | **mIoU** ($\uparrow$)     | **Acc** ($\uparrow$). | **RMSE** ($\downarrow$) | $\delta_1$ ($\uparrow$) |
> | Bilinear          | 35.54                 | 70.20 | 0.510            | 0.818      |
> | Guided Filter     | 32.17                 | 68.25 | 0.537            | 0.802      |
> | FeatUp (JBU)      | 35.60                 | 71.40 | 0.510            | 0.816      |
> | FeatUp (implicit) | 36.70                 | 70.95 | 0.513            | 0.818      |
> | LoftUp            | 3.82                  | 44.35 | 0.534            | 0.803      |
> | JAFAR             | 37.52                 | 72.64 | 0.505            | **0.822**  |
> | **AnyUp (Ours)**  | **37.91**             | **72.78** | **0.502**     | **0.822**  |

---

> ### Author Response · Authors · 2025-11-24
>
> > **Flop / Peak Memory Comparisons.**
>
> We summarize performance in terms of runtime and peak memory usage in the table below (measured on a single NVIDIA A100 GPU).
> Global attention requires $H \times W \times h \times w$ computations, while our window attention, using windows sized $\sigma$ (relative to the input size), significantly reduces the requirement to only $H \times W \times \sigma h \times \sigma w$ computations, which is reflected in both the computational runtime, as well as the peak memory usage.
> In practice, we base our implementation on the NATTEN library [1], which implements CUDA kernels for several architectures that make use of the sparsity in window attention and set $\sigma=0.2$.
> We further extend our code to enable trade-off between memory and speed through chunked computation of the attention that enables even larger input images and feature maps (not used in the benchmarking below).
>
> [1] Hassani, Ali, et al. "Generalized Neighborhood Attention: Multi-dimensional Sparse Attention at the Speed of Light." arXiv preprint arXiv:2504.16922 (2025).
>
> **Table 2: Parameter count and inference time / FLOPs comparison by resolution**
>
> |            | **Resolution**   | **112²**      | **112²**   | **224²**      | **224²**   | **448²**      | **448²**   |
> |------------|------------------|---------------|------------|---------------|------------|---------------|------------|
> | **Model**  | **# Params (M)** | **Time (ms)** | **GFLOPs** | **Time (ms)** | **GFLOPs** | **Time (ms)** | **GFLOPs** |
> | **FeatUp** | 0.2          | 8.6           | **2.6**    | 20.6          | **10.4**   | **73.7**      | **41.6**   |
> | **LoftUp** | 4.3              | 8.2           | 47.1       | 33.5          | 204.1      | 201.2         | 1065.3     |
> | **JAFAR**  | 0.7| $\underline{4.3}$ | 5.5    | $\underline{19.6}$ | 26.8   | 113.9         | 186.1      |
> | **AnyUp (Ours)**   | 0.8      | **2.7**       | $\underline{5.2}$ | **12.8**  | $\underline{20.6}$ | $\underline{92.4}$ | $\underline{82.4}$ |
>
> We can see that in terms of runtime and FLOPS, AnyUp is quite efficient and faster than all previous approaches except FeatUp, which is slightly faster on some resolutions.
>
> **Table 3: Memory usage (GB) by resolution for forward / backward passes**
>
> | **Resolution** | **112²**              | **112²**          | **224²**          | **224²**          | **448²**          | **448²**          |
> |----------------|-----------------------|-------------------|-------------------|-------------------|-------------------|-------------------|
> | **Model**      | **Fwd**               | **Bwd**           | **Fwd**           | **Bwd**           | **Fwd**           | **Bwd**           |
> | **FeatUp**     | $\underline{0.3}$     | **0.4**           | 0.9               | **1.4**           | $\underline{3.4}$ | **5.6**           |
> | **LoftUp**     | **0.2**               | 0.7               | $\underline{0.8}$ | 2.8               | 7.9               | 21.1              |
> | **JAFAR**      | **0.2**               | $\underline{0.5}$ | **0.6**           | $\underline{2.7}$ | 6.9               | 22.3              |
> | **AnyUp (Ours)**       | **0.2**       | 0.8               | $\underline{0.8}$ | 3.3               | **3.3**           | $\underline{12.9}$ |
>
> Similarly to runtime, AnyUp's memory consumption is lower than that of LoftUp and JAFAR and slightly higher than FeatUp's for some resolutions.

---

> ### Author Response · Authors · 2025-11-24
>
> > **Training AnyUp jointly with and within another specialized architecture.**
>
> Thank you for this suggestion, this would indeed be an interesting experiment. It is currently unclear to us which setup, task, and architecture would be specifically of interest or feasible to evaluate in such a way. If you have any specific setup in mind, we would welcome any more specific literature pointers.
> Due to the limited time for rebuttal, we refrain from performing such analysis for now.
>
>
> > **How does AnyUp handle tokens that are non-spatial like those in DINOv2 before the paper Vision Transformers need registers?**
>
> Through its training on local image crops, AnyUp implicitly learns to denoise artifact tokens stemming from ViT attention sinks or positional encodings. This is generally beneficial for better performance on dense tasks [1].
> However, we are aware that some works, especially in the VLM space [2], have found these outlier tokens to be useful in downstream reasoning.
> Thus, if one is explicitly interested in the information stored in these tokens, we recommend using a technique as the one proposed in [3] to identify the attention sinks and extract this information before feature upsampling.
>
> [1] Yang, Jiawei, et al. "Denoising vision transformers." European Conference on Computer Vision. Cham: Springer Nature Switzerland, 2024.
>
> [2] Yu, Zhongzhi, et al. "Unveiling and harnessing hidden attention sinks: enhancing large language models without training through attention calibration." Proceedings of the 41st International Conference on Machine Learning. 2024.
>
> [3] Jiang, Nick, et al. "Vision Transformers Don't Need Trained Registers." Advances in Neural Information Processing Systems, 2025.

---

> ### Author Response · Authors · 2025-11-24
>
> >  **To what extent does this reliance on image parts [during training] change its semantics?**
>
> Thanks for raising this interesting question!
> We agree that the proposed data sampling relies on a certain level of invariance of the feature extractor regarding the global context.
> However, we would like to note that in practice, we sample 224x224 crops in a 448x448 image and, thus, never allow this local crop to be too limited in terms of context.
> We argue that the assumption of invariant representations is reasonable for these crop sizes.
>
> > **What does AnyUp learn that enables such promising transfer? What do these filters learn?**
>
> As can be seen in Tab. 7 of the main paper, an upsampler (using window attention to upsample and trained using our proposed training scheme) that only “sees” the high-resolution guidance image and its downsampled equivalent ($M=0$) is already a quite capable system that is able to upsample features fairly well. However, when not using our feature-agnostic layer, its performance still lags behind the previous state of the art backbone-specific upsampling methods. We hypothesize that this is caused by the fact that a successful upsampler also needs to be able to see the low resolution feature map to distinguish between image areas that are of similar color but of different semantics. When increasing the number of basis filters in our feature-agnostic layer (see Table below), we observe that the upsampling performance increases, which validates our hypothesis. While the learned patterns in the filter basis are not directly interpretable qualitatively, AnyUp using the feature-agnostic layer quantitatively outperforms a “feature-blind” comparable version across all tasks.
>
> **Table 4: Impact of filter basis size on performance.**
>
> |      | Semantic Segmentation |                  |      Depth Estimation     |                           |
> |:----:|:---------------------:|:----------------:|:-------------------------:|:-------------------------:|
> |   **M**  |   **mIoU** ($\uparrow$)   | **Acc** ($\uparrow$) | **RMSE (abs)** ($\downarrow$) | **RMSE (rel)** ($\downarrow$) |
> |  256 |         **62.19**     |     **81.38**    |           0.4765          |         **0.3371**        |
> |  128 |     $\underline{62.16}$ | $\underline{81.37}$ |       **0.4755**      |     $\underline{0.3378}$  |
> |  64  |         62.09         |       81.34      |   $\underline{0.4759}$    |           0.3380          |
> |  32  |         62.07         |       81.31      |           0.4779          |           0.3402          |
> |  16  |         62.09         |       81.32      |           0.4773          |           0.3404          |
> |   8  |         62.00         |       81.27      |           0.4792          |           0.3415          |
> |   0  |         61.97         |       81.23      |           0.4791          |           0.3441          |
> | SOTA |         62.15         |       81.32      |           0.4816          |           0.3413          |

---

### Official Review · Reviewer_kTsf · 2025-11-01

**Soundness:** 3
**Presentation:** 3
**Contribution:** 3
**Rating:** 6
**Confidence:** 5

**Summary:**

Previous feature upsampling methods cannot generalize among different feature encoders, and need to be re-trained for different encoders. This paper proposes AnyUp, which addresses the limitation of existing upsamplers, which is feature-agnostic at inference time. Experiments validate the effectiveness of AnyUp.

**Strengths:**

- The feature-agnostic inference is meaningful for feature upsampling
- The paper is well-written, and the method is simple and easy to follow
- The experiments are comprehensive demostrating the effectiveness. The visualizations are sufficient to explain the design and show the good visual appreance of the generated feature.

**Weaknesses:**

- The techniques used in the method is not very new, for example, window attention-based upsampling has already been used in SAPA.

**Questions:**

- Have you tried training AnyUp on a mixture of features from different encoders (e.g., DINOv2, CLIP, MAE) simultaneously? Would this "universal" training further improve its generalization to completely unseen encoders compared to training on just one?
- You motivate window attention by noting that global attention can pull in "vastly unrelated and distant" features. Can you quantify the performance gain and efficiency (FLOPs/latency) improvement from using windowed attention versus the global attention used in the JAFAR baseline?

---

> ### Author Response · Authors · 2025-11-24
>
> We thank the reviewer for the positive feedback, acknowledging that AnyUp is a **meaningful contribution for feature upsampling**, that the **paper is well-written** and the **experiments are comprehensive**.
>
> > **The techniques used in the method are not very new, see SAPA.**
>
> We appreciate the reviewer’s observation. While window attention-based upsampling has indeed been explored in SAPA, our work differs fundamentally in both motivation and implementation. Our core contribution is the introduction of a first-of-its-kind *learnable, feature-agnostic upsampling model*. Specifically, we adapt and combine a novel feature-agnostic layer, window attention-based upsampling, and a local crop-based training strategy, resulting in state-of-the-art performance on feature upsampling *while* being the first approach applicable to any input feature.
>
> > **Training on a mixture of features from different encoders.**
>
> Thank you for the suggestion. We performed an experiment where we jointly trained AnyUp on DINOv2, SigLIP 2, and CLIP features. The results for applying it to different features are shown below. We can see that, on average, the model trained on all features clearly outperforms the model only trained on DINOv2. For testing on DINOv2 specifically, the DINOv2-trained model still performs slightly better. The difference is marginal though. More importantly, the model trained on multiple features generalizes better to completely unseen features, i.e., the DeiT features [1]. We will include these results in the updated paper.
>
> [1] Touvron, Hugo, Matthieu Cord, and Hervé Jégou. "DeiT III: Revenge of the ViT." European Conference on Computer Vision. Cham: Springer Nature Switzerland, 2022.
>
> **Table 1: Generalization of AnyUp trained on multiple feature encoders.**
>
> |                                                | DINOv2 ViT-S |       |            |            | SigLIP 2 ViT-B |       |            |            | DeiT  |       |            |            | Average |       |            |            |
> |------------------------------------------------|--------------|-------|------------|------------|----------------|-------|------------|------------|-------|-------|------------|------------|---------|-------|------------|------------|
> | AnyUp trained on:                              | mIoU ($\uparrow$) | Acc ($\uparrow$) | RMSE (abs) ($\downarrow$) | RMSE (rel) ($\downarrow$) | mIoU ($\uparrow$) | Acc ($\uparrow$) | RMSE (abs) ($\downarrow$) | RMSE (rel) ($\downarrow$) | mIoU ($\uparrow$) | Acc ($\uparrow$) | RMSE (abs) ($\downarrow$) | RMSE (rel) ($\downarrow$) | mIoU ($\uparrow$) | Acc ($\uparrow$) | RMSE (abs) ($\downarrow$) | RMSE (rel) ($\downarrow$) |
> | DINOv2 ViT-S | **62.16** | **81.37** | **0.4755** | **0.3378** | 58.51 | 78.36 | 0.9111 | 0.5780 | 53.93 | 75.90 | 0.6467 | 0.4970 | 58.20 | 78.54 | 0.6778 | 0.4709 |
> | Multi Backbone (incl. DINOv2 ViT-S and SigLIP 2 ViT-B) | 62.04 | 81.30 | 0.4767 | 0.3392 | **59.52** | **79.07** | **0.9088** | **0.5776** | **54.83** | **76.56** | **0.6418** | **0.4886** | **58.80** | **78.98** | **0.6758** | **0.4685** |

---

> ### Author Response · Authors · 2025-11-24
>
> > **Performance gain and efficiency of using window attention?**
>
> We summarize performance in terms of runtime, FLOPs and peak memory usage in the table below (measured on a single NVIDIA A100 GPU).
> Global attention requires $H \times W \times h \times w$ computations, while our window attention, using windows sized $\sigma$ (relative to the input size), significantly reduces the requirement to only $H \times W \times \sigma h \times \sigma w$ computations, which is reflected in both the computational runtime, as well as the peak memory usage.
> In practice, we base our implementation on the NATTEN library [1], which implements CUDA kernels for several architectures that make use of the sparsity in window attention and set $\sigma=0.2$.
> We further extend our code to enable trade-off between memory and speed through chunked computation of the attention that enables even larger input images and feature maps (not used in the benchmarking below).
> As can be seen in Tab. 7 in the main paper, using the proposed window attention instead of global attention for upsampling results in improved performance on downstream tasks.
>
> [1] Hassani, Ali, et al. "Generalized Neighborhood Attention: Multi-dimensional Sparse Attention at the Speed of Light." arXiv preprint arXiv:2504.16922 (2025).
>
> **Table 2: Parameter count and inference time / FLOPs comparison by resolution**
>
> |            | **Resolution**   | **112²**      | **112²**   | **224²**      | **224²**   | **448²**      | **448²**   |
> |------------|------------------|---------------|------------|---------------|------------|---------------|------------|
> | **Model**  | **# Params (M)** | **Time (ms)** | **GFLOPs** | **Time (ms)** | **GFLOPs** | **Time (ms)** | **GFLOPs** |
> | **FeatUp** | 0.2          | 8.6           | **2.6**    | 20.6          | **10.4**   | **73.7**      | **41.6**   |
> | **LoftUp** | 4.3              | 8.2           | 47.1       | 33.5          | 204.1      | 201.2         | 1065.3     |
> | **JAFAR**  | 0.7| $\underline{4.3}$ | 5.5    | $\underline{19.6}$ | 26.8   | 113.9         | 186.1      |
> | **AnyUp (Ours)**   | 0.8      | **2.7**       | $\underline{5.2}$ | **12.8**  | $\underline{20.6}$ | $\underline{92.4}$ | $\underline{82.4}$ |
>
> We can see that in terms of runtime and FLOPS, AnyUp is quite efficient and faster than all previous approaches except FeatUp, which is slightly faster on some resolutions.
>
> **Table 3: Memory usage (GB) by resolution for forward / backward passes**
>
> | **Resolution** | **112²**              | **112²**          | **224²**          | **224²**          | **448²**          | **448²**          |
> |----------------|-----------------------|-------------------|-------------------|-------------------|-------------------|-------------------|
> | **Model**      | **Fwd**               | **Bwd**           | **Fwd**           | **Bwd**           | **Fwd**           | **Bwd**           |
> | **FeatUp**     | $\underline{0.3}$     | **0.4**           | 0.9               | **1.4**           | $\underline{3.4}$ | **5.6**           |
> | **LoftUp**     | **0.2**               | 0.7               | $\underline{0.8}$ | 2.8               | 7.9               | 21.1              |
> | **JAFAR**      | **0.2**               | $\underline{0.5}$ | **0.6**           | $\underline{2.7}$ | 6.9               | 22.3              |
> | **AnyUp (Ours)**       | **0.2**       | 0.8               | $\underline{0.8}$ | 3.3               | **3.3**           | $\underline{12.9}$ |
>
> Similarly to runtime, AnyUp's memory consumption is lower than that of LoftUp and JAFAR and slightly higher than FeatUp's for some resolutions.

---

> > ### Comment · Reviewer_kTsf · 2025-11-28
> >
> > Thank the authors for providing a comprehensive rebuttal, which addressed my questions. Due to the simple, effective and universal nature, I' ve decided to keep my positive rating.

---

### Public Comment · ~Minseok_Seo1 · 2025-11-24

I have hands-on experience using AnyUp, and one thing I consistently observe is its strong robustness to noise.

For example, in low-SNR scenarios, AnyUp maintains its performance significantly longer compared to other upsamplers such as FeatUp or LoftUp.

This paper is excellent, and I thought it might be helpful for the authors to be aware of this advantage, so I wanted to leave this comment.

---

### Author Response · Authors · 2025-12-01

We thank the reviewers for their unanimous support and recognition of our work’s contribution to feature upsampling - reflected in the strong initial scores - highlighting three key strengths:

**High Practical Value**: Reviewers commend the clear motivation in solving the “encoder-specific [...] pain point” [9Srj], noting that our “train once, use anywhere” approach [9Srj] is “meaningful for feature upsampling” [kTsf], makes a “great contribution to the community” [spdT] and is of “interest to the community” [MPXT].

**SOTA Performance & Robustness**: The experiments are recognized as “comprehensive” [kTsf] and a “nice panoply” [MPXT], demonstrating that AnyUp achieves “SOTA performance” [9Srj, spdT] while successfully proving its generalization capabilities [9Srj, MPXT].

**Simplicity & Elegance**: The proposed feature-agnostic layer is described as “novel, concise and effective" [spdT]. Reviewers praise the method for being “simple and elegant” [MPXT] and the paper for being “well-written” [kTsf, MPXT, spdT].


**Rebuttal**: The insightful remarks and questions by the reviewers helped us to further improve our paper during the rebuttal:
- We demonstrated that our window attention-based upsampler is **computationally efficient**, halving the amount of FLOPs [JAFAR: 186.1 $\rightarrow$ AnyUp: 82.4 GFLOPs] and memory [JAFAR: 6.9 $\rightarrow$ AnyUp: 3.3 GB] necessary for upsampling to a resolution of 448px compared to prior state-of-the-art upsamplers.
- We further investigated multi-backbone training, possible through AnyUp’s first-of-its-kind feature agnostic nature, and found that training on multiple feature backbones further improves the **generalization to completely unseen features**.
- Finally, we provided **further insights into what makes AnyUp generalize well** by performing additional ablations on our proposed method. In our experiments, we showed that our proposed feature-agnostic layer is crucial to help AnyUp distinguish between areas of similar color but different semantics.

---

### Meta-Review · Area_Chair_JBEP · 2026-01-06

**Summary:**

This paper introduces AnyUp, a universal feature upsampling model that can be trained once and applied to any vision encoder without retraining. All 3 reviewers initially scored 6, with one scored 8. Reviewers appreciated the clear motivation addressing the encoder-specific retraining problem and the feature-agnostic layer design that enables cross-encoder generalization. The authors provided thorough rebuttal experiments including efficiency analysis showing reduced FLOPs compared to JAFAR, cross-encoder generalization tests demonstrating superior performance when applying DINOv2-trained models to SigLIP features, and ablations on the feature-agnostic layer's role. While some concerns were raised about technical novelty given that window attention has been explored in prior work like SAPA, the combination and application to the feature upsampling problem represents meaningful practical contribution. We encourage the authors to release code as promised. Based on the positive reception and strong rebuttal, I recommend accepting this submission.

**Reviewer Concerns:**

Addressed: Authors provided efficiency comparisons (lower FLOPs than JAFAR), cross-encoder generalization experiments (DINOv2→SigLIP transfer outperforms baseline), and feature-agnostic layer ablations demonstrating its importance for generalization. Code release was promised.

Outstanding: Technical novelty of individual components (window attention exists in prior work like SAPA) remains a limitation, though the overall framework provides clear practical value.

**Reviewer Scores:**

GNir (6): Would maintain 6; positive throughout

kTsf (6): Would maintain 6; comprehensive rebuttal addressed questions

X6VJ (6): Would maintain 6; no post-rebuttal engagement

MPXT (8): Would maintain 8 after comprehensive rebuttal

---

### Decision · Program_Chairs · 2026-01-26

Accept (Oral)